# Generative Modelling of Structurally Constrained Graphs

**Manuel Madeira**
EPFL, Lausanne, Switzerland
`manuel.madeira@epfl.ch`

**Clément Vignac**
EPFL, Lausanne, Switzerland

**Dorina Thanou**
EPFL, Lausanne, Switzerland

**Pascal Frossard**
EPFL, Lausanne, Switzerland

## Abstract

Graph diffusion models have emerged as state-of-the-art techniques in graph generation; yet, integrating domain knowledge into these models remains challenging. Domain knowledge is particularly important in real-world scenarios, where invalid generated graphs hinder deployment in practical applications. Unconstrained and conditioned graph diffusion models fail to guarantee such domain-specific structural properties. We present ConStruct, a novel framework that enables graph diffusion models to incorporate hard constraints on specific properties, such as planarity or acyclicity. Our approach ensures that the sampled graphs remain within the domain of graphs that satisfy the specified property throughout the entire trajectory in both the forward and reverse processes. This is achieved by introducing an edge-absorbing noise model and a new projector operator. ConStruct demonstrates versatility across several structural and edge-deletion invariant constraints and achieves state-of-the-art performance for both synthetic benchmarks and attributed real-world datasets. For example, by incorporating planarity constraints in digital pathology graph datasets, the proposed method outperforms existing baselines, improving data validity by up to 71.1 percentage points.

## 1 Introduction

Learning how to generate realistic graphs that faithfully mirror a target distribution is crucial for tasks such as data augmentation in network analysis or discovery of novel network structures. This has become a prominent problem in diverse real-world modelling scenarios, ranging from molecule design [55] and inverse protein folding [86] to anti-money laundering [45] or combinatorial optimization [76]. While the explicit representation of relational and structural information with graphs encourage their widespread adoption in numerous applications, their sparse and unordered nature make the task of graph generation challenging.

In many real-world problems, we possess a priori knowledge about specific properties of the target distribution of graphs. Incorporating such knowledge into generative models is a natural approach to enforce the generated graphs to comply with the domain-specific properties. Indeed, common generative models, even when conditioned towards graph desired properties, fail to offer guarantees. This may however become particularly critical in settings where noncompliant graphs can lead to real-world application failures. Many of these desired properties are edge-related, i.e., constraints in the structure of the graph. For example, in digital pathology, graphs extracted from tissue slides are planar [26, 69]. Similarly, in contact networks between patients and healthcare workers within hospitals, the degrees of healthcare workers are upper bounded to effectively prevent the emergence of superspreaders and mitigate the risk of infectious disease outbreaks [32, 1]. In graph generation,

38th Conference on Neural Information Processing Systems (NeurIPS 2024).

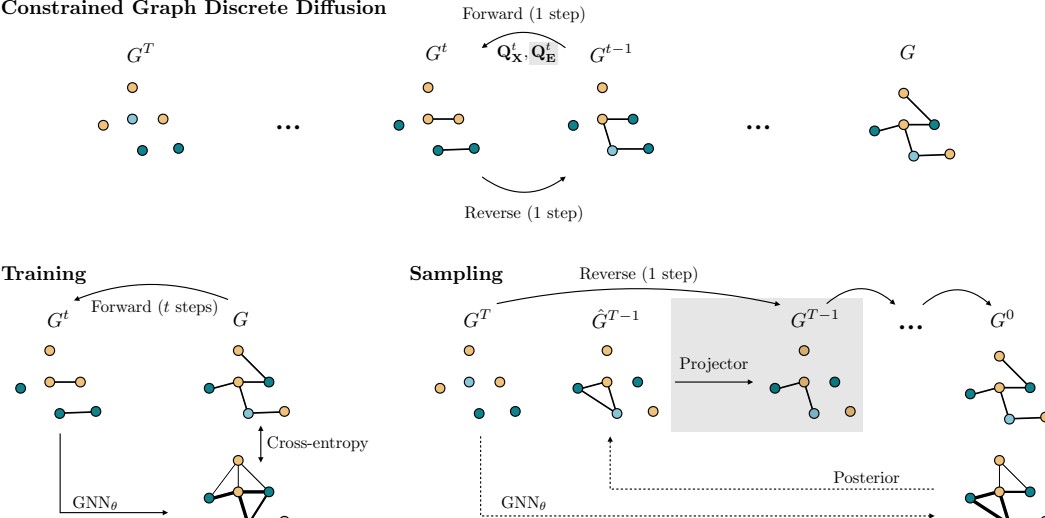

Figure 1: Constrained graph discrete diffusion framework. The forward process consists of an edge deletion process driven by the edge-absorbing noise model, while the node types may switch according to the marginal noise model. At sampling time, the projector operator ensures that sampled graphs remain within the constrained domain throughout the entire reverse process. In the illustrated example, the constrained domain consists exclusively of graphs with no cycles. We highlight in gray the components responsible for preserving the constraining property.

diffusion models have led to state-of-the-art performance [79, 64, 7], in line with their success on other data modalities [75, 29]. However, constrained generation still lags behind its unconstrained counterpart: despite the remarkable expressivity of graph diffusion models, constraining them to leverage specific graph properties remains a particularly challenging task [41].

In this paper, we propose ConStruct, a constrained graph discrete diffusion framework that induces specific structural properties in generative models. Our focus lies on a broad family of structural properties that hold upon edge deletion, including graph planarity or absence of cycles or triangles, for example. ConStruct operates within graph discrete diffusion, where both node and edge types lie in discrete state-spaces [79, 27, 64, 10]. Notably, ConStruct is designed to preserve both the forward and reverse processes of the diffusion model within distribution with respect to a specified structural property. To accomplish this, we introduce two main components: an edge absorbing noise model and an efficient projector of the target property. The former casts the forward process as an edge deletion process and the reverse process as an edge insertion process. Simultaneously, the projector ensures that the inserted edges in the reverse process, predicted by a trained graph neural network, do not violate the structural property constraints. We theoretically ground the projector design by proving that it can retrieve the optimal graph under a graph edit distance setting. Additionally, we further enhance its efficiency by leveraging incremental constraint satisfaction algorithms, as opposed to their full graph versions, and a blocking edge hash table to avoid duplicate constraint property satisfaction checks. These two components enable a reduction in computational redundancy throughout the reverse process.

We empirically validate the benefit of promoting the match of distributions between the training and generative processes in terms of sample quality and constraint satisfaction on a set of benchmark datasets, outperforming unconstrained methods. We demonstrate the flexibility of ConStruct by testing it with three distinct structural properties constraints: graph planarity, acyclicity and lobster components. To further illustrate the utility of ConStruct to real-world applications, we evaluate the performance of our model in generating biologically meaningful cell-cell interactions, represented through planar cell graphs derived from digital pathology data. We focus on the generation of simple yet medically significant tertiary lymphoid structures [44, 18, 58, 28, 69]. Our experiments demonstrate a significant improvement in cell graph generation with ConStruct compared to unconstrained methods [52], notably achieving an increase of up to 71.1 percentage points in terms of cell graph va-

lidity. These results open new venues for innovative data augmentation techniques and novel instance discovery, addressing a key challenge in digital pathology and real-world applications in general.[1]

## 2 Related Work

By decomposing the graph generation task into multiple denoising steps, graph diffusion models have gained prominence due to their superior generative performance in the class of methods that predict the full adjacency matrix at once (e.g., VAEs [40, 74, 78, 37], GANs [16, 42, 54], and normalizing flows [48, 53, 47, 51]). Diverse diffusion formulations have emerged to address various challenges in the graph setting, encompassing score-based approaches [60, 38, 85] and discrete diffusion [79, 27, 64]. They have also been employed as intermediate steps in specific generative schemes, such as hierarchical generation through iterative local expansion [7].

The explicit incorporation of structural information (beyond local biases typical of GNNs) has been shown to be an important prior for enhancing the expressiveness of one-shot graph generative models. For example, in the GAN setting, SPECTRE [54] conditions on graph spectral properties to capture global structural characteristics and achieve improved generative performance. Graph diffusion models are similarly amenable to conditioning techniques [79, 31], which, despite enabling the guidance of the generation process towards graphs with desired properties, do not guarantee the satisfaction of such properties. In contrast, autoregressive models can ensure constraint satisfaction through validity checks at each iteration, effectively addressing this challenge. Although graph diffusion models can leverage formulations that are invariant to permutations, thus avoiding the sensitivity to node ordering that characterizes autoregressive approaches [87, 46, 14], they still lag behind in ensuring constraint satisfaction.

Previous graph diffusion approaches to address this challenge can be categorized according to the nature of the state spaces they assume. In the continuous case, aligned with successful outcomes in other data modalities [12], PRODIGY [72] offers efficient guidance for pre-trained models by relaxing adjacency matrices and categorical node features into continuous spaces, subsequently finding low-overhead projections onto the constraint-satisfying set at each reverse step. This approach can impose structural and molecular properties for which closed-form projections can be derived. However, it does not guarantee constraint satisfaction, facing a trade-off between performance and constraint satisfaction due to mismatched training and sampling distributions. This challenge arises from the continuous relaxation approach, which, while effective within the plug-and-play controllable diffusion framework, imposes an implicit ordering between states that can yield suboptimal graph representations when remapping to the inherently discrete graph space. Additionally, the proposed projection operators cannot be derived for some combinatorial constraints over the graph structure that are frequently encountered in real-world scenarios, such as planarity and acyclicity.

Then, in discrete state-spaces, EDGE [10] leverages a node-wise maximum degree hard constraint due to its degree guidance but it is limited to this particular property. Similarly, GraphARM [41], a graph autoregressive diffusion model, allows for constraint incorporation in the autoregressive manner. However, this method requires learning a node ordering, a task that is at least as complex as isomorphism testing. Therefore, to the best of our knowledge, ConStruct consists of the first constrained graph discrete diffusion framework covering a broad class of structural (potentially combinatorial) constraints.

## 3 Constrained Graph Diffusion Models

We now introduce our framework on generative modelling for structurally constrained graphs. We first present the graph diffusion framework and then focus on the new components for constrained graph generation.

### 3.1 Graph Diffusion Models

We first introduce the mathematical notation adopted in the paper.

---

[1]Our code and data are available at https://github.com/manuelmlmadeira/ConStruct.

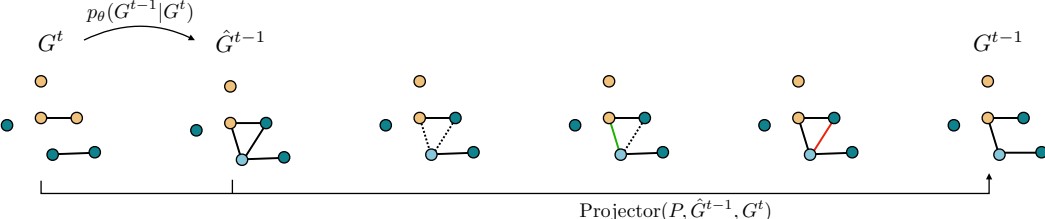

Figure 2: Projector operator. At each iteration, we start by sampling a candidate graph $\hat{G}^{t-1}$ from the distribution $p_\theta(G^{t-1}|G^t)$ provided by the diffusion model. Then, the projector step inserts in an uniformly random manner the candidate edges, discarding those that violate the target property, $P$, i.e., acyclicity in this illustration. In the end of the reverse step, we find a graph $G^{t-1}$ that is guaranteed to comply with such property.

**Notation**   We define a graph as $G = (X, E)$, where $X$ and $E$ denote the sets of attributed nodes and edges, respectively. We consider the node and edge features to be categorical and to lie in the spaces $\mathcal{X}$ and $\mathcal{E}$ of cardinalities $b$ and $c$, respectively. Thus, $x_i$ denotes the node attribute of node $i$ and $e_{ij}$ the edge attribute of the edge between nodes $i$ and $j$. With $\mathcal{H}^k = \{\mathbf{v} = (v_1, \ldots, v_k) \mid v_i \in \{0,1\}, \sum_{i=1}^k v_i = 1\}$, their corresponding one-hot encodings are then $\mathbf{x}_i \in \mathcal{H}^b$ and $\mathbf{e}_{ij} \in \mathcal{H}^{c+1}$, since we consider the absence of edge between two nodes as an edge type ("no edge" type). These are stacked in tensors $\mathbf{X} \in \{0,1\}^{n \times b}$ and $\mathbf{E} \in \{0,1\}^{n \times n \times (c+1)}$, respectively. So, equivalently to the set notation, we also have $G = (\mathbf{X}, \mathbf{E})$. Additionally, we define the probability simplex, $\Delta^k = \{(\lambda_0, \lambda_1, \ldots, \lambda_{k-1}) \in \mathbb{R}^k \mid \lambda_i \geq 0 \text{ for all } i, \ \sum_{i=0}^{k-1} \lambda_i = 1\}$.

We then recall the core components of generative models based on graph diffusion, a state-of-the-art framework in several applications [38, 79]. Graph diffusion models are composed of two main processes: a *forward* and a *reverse* one. The forward process consists of a Markovian noise model, $q$, with $T$ timesteps, that allows to progressively perturb a clean graph $G$ to its noisy version $G^t$, where $t \in \{1, \ldots, T\}$. This process is typically modelled independently for nodes and edges. The reverse process consists of the opposite development, starting from a fully noisy, $G^T$, and iteratively refining it until a new clean sample is generated. This process uses a denoising neural network (NN), the only learnable part of the diffusion model. The NN is trained to predict a probability distribution over node and edge types of the clean graph $G$. After its training, we combine the NN prediction with the posterior term of the forward process to find the distribution $p_\theta(G^{t-1}|G^t)$, from where we sample a one-step denoised graph. The reverse process results from applying this sampling procedure iteratively until we arrive to a fresh new clean graph $G^0$. Both processes are illustrated in Figure 1.

In some tasks, we are interested in generating instances of a specific class of graphs that conform to well-defined structural properties and align with the training distribution. Importantly, these structural properties do not fully define the underlying distribution; rather, the model must still learn this distribution within the specific class of graphs. This approach becomes particularly crucial in scenarios where we possess domain knowledge but lack sufficient data for an unconstrained model to capture strict dependencies, allowing us to reduce the task's hypothesis space. This need also applies to many real-world applications, where generated graphs become irrelevant if they do not meet certain conditions, as they may be infeasible or lack physical meaning (e.g., in drug design). Despite the remarkable expressivity of graph diffusion models, incorporating such constraints into their generative process remains a largely unsolved problem.

## 3.2   Constrained Graph Discrete Diffusion Models

We now introduce ConStruct, a framework that efficiently constrains graph diffusion models based on structural properties. Constraining graph generation implies guaranteeing that such target structural properties are not violated in the generated graphs. We build on graph discrete diffusion due to its intrinsic capability to effectively preserve fundamental structural properties (e.g., sparsity) of graphs throughout the generative process [64, 79, 27].

A successful way of imposing constraints to diffusion models in continuous state-spaces consists of constraining the domain where the forward and reverse processes occur [50, 22, 23]. However,

constraining domains over graphs, which are inherently discrete, poses a challenging combinatorial problem. Instead, we propose to constrain the graph generative process with specific structural properties. In our approach, we explore the broad class of graph structural properties that hold under edge deletion, namely *edge-deletion invariant* properties.

**Definition 3.1.** (Edge-Deletion Invariance) *Let $P$ be a boolean-valued application defined on graphs, referred to as a property. $P$ is said to be edge-deletion invariant if, for any graph $G$ and any subset of edges $\tilde{E} \subset E$, it satisfies:*

$$P(G) = \textit{True} \implies P(G') = \textit{True}, \quad \textit{with } G' = (X, E \setminus \tilde{E}).$$

Many properties that are observed in real-world scenarios are edge-deletion invariant. For example, graph planarity is observed in road networks [83], chip design [8], biochemistry [73] or digital pathology [36]. In evolutionary biology [25] or epidemiology [70], we find graphs that must not have cycles. Additionally, if we consider the extensions of discrete diffusion to directed graphs (e.g., Asthana et al. [3]), there are several domains where graph acyclicity is critical: neural architecture search, bayesian network structure learning [88], or causal discovery [67]. Also, maximum degree constraints are quite common in the design of contact networks [32, 1]. Finally, it is worth noting that Definition 3.1 is extendable to continuous graph-level features through a binary decision (e.g., by thresholding continuous values into boolean values).

Provided that the training graphs satisfy the target structural properties, ConStruct enforces these properties in the generated graphs by relying on two main components: an edge-absorbing noise model and a projector. These two components are described in detail below.

## 3.3 Edge-deletion Aware Forward Process

Our goal is to design a forward process that yields noisy graphs that necessarily satisfy the target property. This process is typically modelled using transition matrices. Thus, $[\mathbf{Q}_X^t]_{ij} = q(x^t = j|x^{t-1} = i)$ corresponds to the probability of a node transitioning from type $i$ to type $j$. Similarly, for edges we have $[\mathbf{Q}_E^t]_{ij} = q(e^t = j|e^{t-1} = i)$. These are applied independently to each node and edge, yielding $q(G^t|G^{t-1}) = (\mathbf{X}^{t-1}\mathbf{Q}_X^t, \mathbf{E}^{t-1}\mathbf{Q}_E^t)$. Consequently, we can directly jump $t$ timesteps in the forward step through the categorical distribution given by:

$$q(G^t|G) = (\mathbf{X}\bar{\mathbf{Q}}_X^t, \mathbf{E}\bar{\mathbf{Q}}_E^t), \tag{1}$$

with $\bar{\mathbf{Q}}_X^t = \mathbf{Q}_X^1 \ldots \mathbf{Q}_X^t$ and $\bar{\mathbf{Q}}_E^t = \mathbf{Q}_E^1 \ldots \mathbf{Q}_E^t$. Noising a graph amounts to sampling a graph from this distribution. For the nodes, we use the marginal noise model [79] due to its great empirical performance. Importantly, to preserve the constraining structural property throughout the forward process, and, consequently, throughout the training algorithm (see Algorithm 1, in Appendix A.1), we propose the utilization of an *edge-absorbing noise model* [4]. This noise model forces each edge to either remain in the same state or to transition to an absorbing state (which we define to be the no-edge state) throughout the forward process. This edge noise model poses the forward as an edge deletion process, converging to a limit distribution that yields graphs without edges. Therefore, we obtain the following transition matrices:

$$\begin{aligned}
\mathbf{Q}_X^t &= \alpha^t \mathbf{I} + (1 - \alpha^t)\mathbf{1}_b \mathbf{m}_X' \quad \text{and} \\
\mathbf{Q}_E^t &= \alpha_{\text{ABS}}^t \mathbf{I} + (1 - \alpha_{\text{ABS}}^t)\mathbf{1}_c \mathbf{e}_E',
\end{aligned} \tag{2}$$

where $\alpha^t$ and $\alpha_{\text{ABS}}^t$ transition from 1 to 0 with $t$ according to the popular cosine scheduling [59] and the mutual-information-based noise schedule ($\alpha^t = 1 - (T + t + 1)^{-1}$) [4], respectively. The vectors $\mathbf{1}_b \in \{1\}^b$ and $\mathbf{1}_c \in \{1\}^{c+1}$ are filled with ones, and $\mathbf{m}_X' \in \Delta^b$ and $\mathbf{e}_E' \in \mathcal{H}^{c+1}$ are row vectors filled with the marginal node distribution and the one-hot encoding of the no-edge state, respectively.

## 3.4 Structurally-Constrained Reverse Process

The reverse process of the diffusion model is fully characterized by the distribution $p_\theta(G^{t-1}|G^t)$. We detail how to build it from the predictions of a denoising graph neural network, $\text{GNN}_\theta$, and the posterior term of the forward process in Appendix A.2. Importantly, the latter imposes the reverse process as an edge insertion process, yet does not necessarily ensure the target structural property. To handle that, we propose an intermediate procedure for each reverse step. Provided a noisy graph $G^t$

at timestep $t$, we do not accept directly $\hat{G}^{t-1}$, sampled from $p_\theta(G^{t-1}|G^t)$, as the one step denoised graph. Instead, we iteratively insert the newly added edges to $\hat{G}^{t-1}$ in a random order, discarding the ones that lead to the violation of the target property. Therefore, we only have $G^{t-1} = \hat{G}^{t-1}$ if none of the candidate edges breaks the target property. We refer to the operator that outputs $G^{t-1}$ provided $\hat{G}^{t-1}$ and $G^t$ by discarding the violating edges as the *projector*. Its implementation is illustrated in Figure 2 and described in Algorithm 2, in Appendix A.3. Importantly, this procedure merely interferes with the sampling algorithm (refer to Algorithm 3, in Appendix A.3) and ensures that the diffusion model training remains unaffected, fully preserving its efficiency.

Despite its algorithmic simplicity, the design of our projector is theoretically motivated by the result below. We denote the graph edit distance [68] with uniform cost between two graphs $G_1$ and $G_2$ by $\text{GED}(G_1, G_2)$ (see Definition B.1).

**Theorem 1.** (Simplified) *Let $\mathcal{G}^{t-1} = \text{Projector}(P, \hat{G}^{t-1}, G^t)$ be the set of all possible one-step denoised graphs outputted by ConStruct. If we define $G^*$ as any optimal solution of:*

$$\min_{G \in \mathcal{C}} \text{GED}(\hat{G}^{t-1}, G), \tag{3}$$

*where $\mathcal{C} = \{G \in \mathcal{G}|P(G) = True, G \supset G^t\}$ and $\mathcal{G}$ is the set of all unattributed graphs, then $G^*$ can be recovered by our projector, i.e., $G^* \in \mathcal{G}^{t-1}$.*

The relationship between the projector, the candidate element $\hat{G}^{t-1}$ (the instance we aim to project onto a constrained set) and the specified target property $P$ (defining the constrained set) can be analogized to the conventional projection operator in continuous state spaces. However, while projection in continuous spaces is typically straightforward, this is not the case for discrete state spaces, where, for instance, there often lacks an inherent notion of order between different states. In particular, projecting into an arbitrary subclass of graphs is a complex general combinatorial problem to which there is no efficient solution. For example, finding the maximum planar subgraph of a given graph is NP-hard [11]. Therefore, the novelty of our method is introduced by considering an additional dependency on $G^t$: to make such problem efficiently approachable, we use the previous iterate, $G^t$, which we know by construction that verifies the target property, as a reference. This information is added into the optimization problem through the formulation of the set $C$. Importantly, this formulation is consistent with the designed noise for the diffusion model, as it complies with the reverse process as an edge insertion process (i.e., $G^t \subset G^{t-1}$). The complete version of this theorem and extensions for specific constraints can be found in Appendix B.

Importantly, the utilization of the projector breaks the independent sampling of new edges since the insertion of an edge now depends on the order by which we insert them at a given timestep. This sacrifices the tractability of an evidence lower bound for the diffusion model's likelihood. In exchange, it conserves all the sampled graphs throughout the reverse process in the constrained domain. Therefore, the edge-absorbing noise model and the projector jointly ensure that the graph distributions of the training and sampling procedures match, within the predefined constrained graph domain. With these blocks in place, we are now able to both train and sample from the constrained diffusion model.

### 3.5 Implementation Improvements

We further enhance the efficiency of the sampling algorithm with the two improvements detailed below.

**Blocking Edge Hash Table**     Throughout the reverse process, we keep in memory the edges that have already been rejected in previous timesteps (higher $t$). Therefore, once an edge is rejected, it is blocked throughout the rest of the reverse process. This prevents the repetition of redundant constraint satisfaction checks since we know *a priori* that inserting a previously rejected edge would lead to constraint violation. We store this information in a hash table, where both the lookup and update operations are $O(1)$, causing minor overhead. Since we only perform the validity check, of complexity $O(V)$, once for each edge - if it is a candidate edge, we either insert it or block it -, it incurs a $O(n^2 V)$ overhead throughout the full reverse process. Note that we lose any dependency on the number of timesteps of the reverse process, which is typically the limiting factor in diffusion models efficiency due to its required high values ($T \approx 10^3$).

**Incremental Algorithms**     Our reverse process consists solely of edge insertion steps, making it well-suited for the application of incremental algorithms. These algorithms efficiently check whether newly

added edges preserve the target property by updating and checking smartly designed representations of the graph. This approach contrasts with full graph counterparts, leading to significant efficiency gains by reducing redundant computation. For instance, while the best full planar testing algorithm is $O(n)$ [30], its fastest known incremental test has amortized running time of $O(\alpha(\mathfrak{q}, n))$, where $\mathfrak{q}$ is the total number of operations (edge queries and insertions), and $\alpha$ denotes the inverse-Ackermann function [43] (often considered "almost constant" complexity). More details for different properties in Appendix C.

At each reverse step, the denoising network makes predictions for all nodes and pairs of nodes. This results in $O(n^2)$ predictions per step. Thus, the complexity of the sampling algorithm of the underlying discrete diffusion model is $O(n^2T)$. In addition, the complexity overhead imposed by the projector is $O(NV)$. Here, $V$ represents the complexity of the property satisfaction algorithm and $N$ is the total number of times this algorithm is applied throughout the reverse process. So, in total, we have $O(n^2T + NV)$. Our analysis in Appendix C shows that incremental property satisfaction algorithms have notably low complexity. For instance, in cases like acyclicity, lobster components, and maximum degree, we have $V = O(|E_{\text{added}}|)$. Since the projector adds one edge at a time, we have $V = O(1)$. Additionally, since the blocking edge hash table limits us to perform at most one property satisfaction check per newly proposed edge (either we have never tested it or it is already blocked), $N$ corresponds to the total number of different edges proposed by the diffusion model across the whole reverse process. A reasonable assumption is that the model proposes $N = O(|E|)$ edges throughout the reverse process, with $|E|$ referring to the number of edges of the clean graph. This is for example true if the model is well trained and predicts the correct graph. Most families of graphs are sparse, meaning that $O(|E|/n^2) \to 0$ as $n \to \infty$. For example, planar and tree graphs can be shown to satisfy $|E|/n^2 = O(1/n)$. Thus, we necessarily have $N \leq n^2$. For these reasons, we directly find $O(NV) \ll O(n^2T)$, highlighting the minimal overhead imposed by the projector compared to the discrete diffusion model. This explains the low runtime overhead observed for ConStruct, as detailed in Appendix D.3 (9% for graphs of the tested size). Therefore, we can conclude that asymptotically $O(n^2T + NV) = O(n^2T)$, i.e., the projector overhead becomes increasingly negligible relative to the diffusion algorithm itself as the graph size increases, highlighting the scalability of our method.

## 4 Experiments

In this section, we first explore the flexibility of ConStruct to accommodate different constraints in synthetic unattributed graph datasets. Then, we test its applicability to a real-world scenario with digital pathology data.

### 4.1 Synthetic Graphs

**Setup** We focus on three synthetic datasets with different structural properties: the *planar* dataset [54], composed of planar and connected graphs; the *tree* dataset [7], composed of connected graphs without cycles (tree graph); and the *lobster* dataset [46], composed of connected graphs without cycles, where no node is more than 2 hops away from a backbone path (lobster graph). We follow the splits originally proposed for each of the datasets: 80% of the graphs are used in the training set and the remaining 20% are allocated to the test set. We use 20% of the train set as validation set. Statistics of these datasets are shown in Appendix E. As the graphs in these datasets are unattributed, we can specifically isolate ConStruct's capability of incorporating structural information in comparison to previously proposed methods, which are described in Appendix E.2. From here on, we use DiGress+ to denote the DiGress model with the added extra features described in Appendix A.1 and HSpectre to refer to the model proposed by Bergmeister et al. [7].

Regarding performance metrics, we follow the evaluation procedures from Martinkus et al. [54]. We assess how close the distributions of different graph statistics computed from the generated and test sets are. To accomplish that, we compute the Maximum Mean Discrepancy (MMD)[2] for the node degrees (Deg.), clustering coefficients (Clus.), orbit count (Orbit), eigenvalues of the normalized graph Laplacian (Spec.), and statistics from a wavelet graph transform (Wavelet). To summarize this set of metrics, we compute the ratios against the corresponding metrics from the training set and then average them (Ratio). We also compute the proportion of generated graphs that are non-isomorphic to each other (Unique), the proportion that are non-isomorphic to any graph in the training set (Novel),

---

[2]To align with previous literature, we actually compute $\text{MMD}^2$.

Table 1: Graph generation performance on synthetic graphs. We present the results over five sampling runs of 100 generated graphs each, in the format mean ± standard error of the mean. The remaining values are retrieved from Bergmeister et al. [7] for the planar and tree datasets, and from Dai et al. [14] and Jang et al. [34] for the lobster dataset. For the average ratio computation, we follow [7] and do not consider the metrics whose train set MMD is 0. We recompute the train set MMDs according to our splits but, for fairness, in the retrieved methods the average ratio metric is not recomputed.

| Model | Deg. ↓ | Clus. ↓ | Orbit ↓ | Spec. ↓ | Wavelet ↓ | Ratio ↓ | Valid ↑ | Unique ↑ | Novel ↑ | V.U.N. ↑ | Property ↑ |
|---|---|---|---|---|---|---|---|---|---|---|---|
| | | | | | Planar Dataset | | | | | | |
| Train set | 0.0002 | 0.0310 | 0.0005 | 0.0038 | 0.0012 | 1.0 | 100 | 100 | 0.0 | 0.0 | 100 |
| GraphRNN [87] | 0.0049 | 0.2779 | 1.2543 | 0.0459 | 0.1034 | 490.2 | 0.0 | 100 | 100 | 0.0 | — |
| GRAN [46] | 0.0007 | 0.0426 | 0.0009 | 0.0075 | 0.0019 | 2.0 | 97.5 | 85.0 | 2.5 | 0.0 | — |
| SPECTRE [54] | 0.0005 | 0.0785 | 0.0012 | 0.0112 | 0.0059 | 3.0 | 25.0 | 100 | 100 | 25.0 | — |
| DiGress [79] | 0.0007 | 0.0780 | 0.0079 | 0.0098 | 0.0031 | 5.1 | 77.5 | 100 | 100 | 77.5 | — |
| EDGE [10] | 0.0761 | 0.3229 | 0.7737 | 0.0957 | 0.3627 | 431.4 | 0.0 | 100 | 100 | 0.0 | — |
| BwR [17] | 0.0231 | 0.2596 | 0.5473 | 0.0444 | 0.1314 | 251.9 | 0.0 | 100 | 100 | 0.0 | — |
| BiGG [14] | 0.0007 | 0.0570 | 0.0367 | 0.0105 | 0.0052 | 16.0 | 62.5 | 85.0 | 42.5 | 5.0 | — |
| GraphGen [24] | 0.0328 | 0.2106 | 0.4236 | 0.0430 | 0.0989 | 210.3 | 7.5 | 100 | 100 | 7.5 | — |
| HSpectre (one-shot) [7] | 0.0003 | 0.0245 | 0.0006 | 0.0104 | 0.0030 | 1.7 | 67.5 | 100 | 100 | 67.5 | — |
| HSpectre [7] | 0.0005 | 0.0626 | 0.0017 | 0.0075 | 0.0013 | 2.1 | 95.0 | 100 | 100 | 95.0 | — |
| DiGress+ | 0.0008 ±0.0001 | 0.0410 ±0.0033 | 0.0048 ±0.0004 | 0.0056 ±0.0004 | 0.0020 ±0.0002 | 3.6 ±0.2 | 76.4 ±1.3 | 100.0 ±0.0 | 100.0 ±0.0 | 76.4 ±1.3 | 76.4 ±1.3 |
| ConStruct | 0.0003 ±0.0001 | 0.0403 ±0.0047 | 0.0004 ±0.0001 | 0.0053 ±0.0004 | 0.0009 ±0.0001 | 1.1 ±0.1 | 100.0 ±0.0 | 100.0 ±0.0 | 100.0 ±0.0 | **100.0** ±0.0 | 100.0 ±0.0 |
| | | | | | Tree Dataset | | | | | | |
| Train set | 0.0001 | 0.0000 | 0.0000 | 0.0075 | 0.0030 | 1.0 | 100 | 100 | 0.0 | 0.0 | 100 |
| GRAN [46] | 0.1884 | 0.0080 | 0.0199 | 0.2751 | 0.3274 | 607.0 | 0.0 | 100 | 100 | 0.0 | — |
| DiGress [79] | 0.0002 | 0.0000 | 0.0000 | 0.0113 | 0.0043 | 1.6 | 90.0 | 100 | 100 | 90.0 | — |
| EDGE [10] | 0.2678 | 0.0000 | 0.7357 | 0.2247 | 0.4230 | 850.7 | 0.0 | 7.5 | 100 | 0.0 | — |
| BwR [17] | 0.0016 | 0.1239 | 0.0003 | 0.0480 | 0.0388 | 11.4 | 0.0 | 100 | 100 | 0.0 | — |
| BiGG [14] | 0.0014 | 0.0000 | 0.0000 | 0.0119 | 0.0058 | 5.2 | 100 | 87.5 | 50.0 | 75.0 | — |
| GraphGen [24] | 0.0105 | 0.0000 | 0.0000 | 0.0153 | 0.0122 | 33.2 | 95.0 | 100 | 100 | 95.0 | — |
| HSpectre (one-shot) [7] | 0.0004 | 0.0000 | 0.0000 | 0.0080 | 0.0055 | 2.1 | 82.5 | 100 | 100 | 82.5 | — |
| HSpectre [7] | 0.0001 | 0.0000 | 0.0000 | 0.0117 | 0.0047 | 4.0 | 100 | 100 | 100 | **100** | — |
| DiGress+ | 0.0002 ±0.0001 | 0.0000 ±0.0000 | 0.0000 ±0.0000 | 0.0092 ±0.0005 | 0.0032 ±0.0001 | 1.3 ±0.2 | 91.6 ±0.7 | 100.0 ±0.0 | 100.0 ±0.0 | 91.6 ±0.7 | 97.0 ±0.8 |
| ConStruct | 0.0003 ±0.0001 | 0.0000 ±0.0000 | 0.0000 ±0.0000 | 0.0073 ±0.0008 | 0.0034 ±0.0002 | 1.9 ±0.3 | 83.0 ±1.8 | 100.0 ±0.0 | 100.0 ±0.0 | 83.0 ±1.8 | 100.0 ±0.0 |
| | | | | | Lobster Dataset | | | | | | |
| Train set | 0.0002 | 0.0000 | 0.0000 | 0.0070 | 0.0070 | 1.0 | 100 | 100 | 0.0 | 0.0 | 100 |
| GraphRNN [87] | 0.000 | 0.000 | 0.000 | 0.011 | — | — | 100 | — | — | — | — |
| GRAN [46] | 0.038 | 0.000 | 0.001 | 0.027 | — | — | 88.0 | — | — | — | — |
| GraphGen [24] | 0.548 | 0.040 | 0.247 | — | — | — | — | — | — | — | — |
| GraphGen-Redux [5] | 1.189 | 1.859 | 0.885 | — | — | — | — | — | — | — | — |
| BiGG [14] | 0.000 | 0.000 | 0.000 | 0.009 | — | — | 100 | — | — | — | — |
| GDSS [38] | 0.117 | 0.002 | 0.149 | — | — | — | 18.2 | 100 | 100 | 18.2 | — |
| BwR [17] | 0.316 | 0.000 | 0.247 | — | — | — | 100 | 63.6 | 100 | 63.6 | — |
| GEEL [34] | 0.002 | 0.000 | 0.001 | — | — | — | 72.7 | 100 | 72.7 | ≤ 72.7 | — |
| HGGT [33] | 0.003 | 0.000 | 0.015 | — | — | — | — | — | — | — | — |
| DiGress [79] | 0.021 | 0.000 | 0.004 | — | — | — | 54.5 | 100 | 100 | 54.5 | — |
| DiGress+ | 0.0005 ±0.0001 | 0.0000 ±0.0000 | 0.0000 ±0.0000 | 0.0114 ±0.0006 | 0.0093 ±0.0005 | 1.8 ±0.1 | 79.0 ±1.1 | 98.0 ±0.7 | 96.6 ±0.6 | 69.4 ±1.2 | 76.8 ±1.7 |
| ConStruct | 0.0003 ±0.0001 | 0.0000 ±0.0000 | 0.0000 ±0.0000 | 0.0092 ±0.0009 | 0.0074 ±0.0004 | 1.3 ±0.2 | 86.8 ±2.4 | 98.8 ±0.6 | 97.0 ±0.9 | **83.2** ±2.3 | 100.0 ±0.0 |

and the proportion of generated graphs that are valid (Valid). Graphs are considered valid if they are planar and connected, trees, or lobster graphs, when the generative model is trained on the planar, tree, or lobster dataset, respectively. We merge these three metrics through the proportion of generated graphs that are simultaneously valid, unique and novel (V.U.N.).

**Constraining Criteria**  Various constraining criteria are chosen for ConStruct according to the structural properties of each dataset. For the planar dataset, we use planarity. For the tree dataset, we impose the absence of cycles. For the lobster dataset, we constrain the graph domain to those graphs whose connected components are lobsters. To check to what extent these criteria are verified by the compared methods, we compute the proportion of generated graphs that comply with the selected constraining criterion of the corresponding dataset (listed under the "Property" column in Table 1).

**Graph Generation Performance**  We present the results in Table 1. For the planar dataset, ConStruct achieves nearly optimal performance, clearly outperforming all other methods. It is actually the first method to achieve 100% V.U.N., indicating state-of-the-art performance. Moreover, in terms of average ratio, it clearly outperforms all other methods, with the average ratio approaching 1, suggesting high sample quality. Regarding the lobster dataset, ConStruct exhibits a similar trend, demonstrating superior performance compared to DiGress+. It leads to state-of-the-art results in both average ratio and V.U.N. metrics. The lower novelty and uniqueness values (<100%) are attributed to the dataset's smaller size. In fact, we train both models on 64 examples (80% of the train set) while generating 100 graphs in each run. Conversely, for the tree dataset, ConStruct is outperformed by DiGress+ due to the marginally lower expressivity of the edge-absorbing noise model for this particular case (see Appendix H.1 for details). As a sanity check, we observe that for all three datasets, ConStruct ensures the constraining property for all generated graphs. However, the validity values are below 100% (except for planar) since connectedness of the generated graph is not guaranteed. In

Table 2: Graph generation performance on digital pathology graphs. We present the results for each method over five sampling runs of 100 generated graphs each, in the format mean ± standard error of the mean.

| | Low TLS Dataset | | | | | | | | | | |
|---|---|---|---|---|---|---|---|---|---|---|---|
| Model | Ratio↓ | Conn.↑ | Planar↑ | V.U.N.↑ | $\kappa(0)$↓ | $\kappa(1)$↓ | $\kappa(2)$↓ | $\kappa(3)$↓ | $\kappa(4)$↓ | $\kappa(5)$↓ | TLS Valid↑ |
| Train set | 1.0 | 100 | 100 | 0.0 | 0.6928 | 0.0000 | 0.0000 | 0.0000 | 0.0000 | 0.0000 | 100 |
| Baseline [52] | 194.6 ±3.0 | 50.3 ±0.7 | 10.0 ±0.5 | 0.0 ±0.0 | 0.6256 ±0.0228 | 0.2350 ±0.0228 | 0.2350 ±0.0000 | 0.0470 ±0.0470 | 0.0000 ±0.0000 | 0.0000 ±0.0000 | 0.0 ±0.0 |
| GraphGen [24] | 212.7 ±4.2 | 100.0 ±0.0 | 33.3 ±0.5 | 33.0 ±1.8 | 0.7354 ±0.0220 | 0.1880 ±0.0470 | 0.0470 ±0.0470 | 0.0000 ±0.0000 | 0.0000 ±0.0000 | 0.0000 ±0.0000 | 33.3 ±0.5 |
| BiGG [14] | 132.0 ±6.3 | 99.5 ±0.1 | 23.3 ±0.6 | 0.8 ±0.2 | 0.6184 ±0.0437 | 0.1410 ±0.0576 | 0.0470 ±0.0470 | 0.0470 ±0.0470 | 0.0470 ±0.0470 | 0.0470 ±0.0470 | 23.3 ±0.6 |
| SPECTRE [54] | 427.0 ±4.3 | 95.3 ±0.2 | 51.2 ±0.6 | 15.8 ±1.2 | 0.2350 ±0.0000 | 0.0000 ±0.0000 | 0.0000 ±0.0000 | 0.0000 ±0.0000 | 0.0000 ±0.0000 | 0.0000 ±0.0000 | 50.6 ±0.7 |
| DiGress+ | **4.9** ±1.0 | 96.0 ±0.7 | 19.8 ±1.8 | 18.6 ±1.8 | 0.7306 ±0.0371 | 0.1410 ±0.0576 | 0.0000 ±0.0000 | 0.0000 ±0.0000 | 0.0000 ±0.0000 | 0.0000 ±0.0000 | 18.6 ±1.8 |
| ConStruct | 4.4 ±0.3 | 98.4 ±0.8 | 100.0 ±0.0 | 98.4 ±0.8 | 0.6781 ±0.0795 | 0.2350 ±0.0000 | 0.0940 ±0.0576 | 0.0000 ±0.0000 | 0.0000 ±0.0000 | 0.0000 ±0.0000 | 96.2 ±0.7 |
| | High TLS Dataset | | | | | | | | | | |
| Train set | 1.0 | 100 | 100 | 0.0 | 0.4257 | 0.4512 | 0.4745 | 0.6395 | 0.7770 | 0.7663 | 100 |
| Baseline [52] | 354.9 ±2.2 | 49.8 ±0.3 | 3.4 ±0.2 | 0.2 ±0.2 | 0.3276 ±0.0023 | 0.3412 ±0.0070 | 0.3669 ±0.0172 | 0.5096 ±0.0157 | 0.6231 ±0.0176 | 0.6988 ±0.0203 | 0.0 ±0.0 |
| GraphGen [24] | 559.1 ±9.8 | 100.0 ±0.0 | 48.1 ±0.6 | 47.4 ±1.6 | 0.3311 ±0.0158 | 0.3620 ±0.0228 | 0.4613 ±0.0121 | 0.6034 ±0.0359 | 0.7500 ±0.0231 | 0.7523 ±0.0346 | 16.9 ±0.6 |
| BiGG [14] | 307.7 ±15.5 | 99.5 ±0.1 | 10.1 ±0.8 | 0.4 ±0.2 | 0.3706 ±0.0225 | 0.4850 ±0.0361 | 0.5970 ±0.0166 | 0.7151 ±0.0112 | 0.7494 ±0.0128 | 0.7515 ±0.0220 | 10.0 ±0.8 |
| SPECTRE [54] | 938.1 ±4.1 | 91.3 ±0.3 | 0.0 ±0.0 | 0.0 ±0.0 | 0.3190 ±0.0293 | 0.3585 ±0.0279 | 0.4033 ±0.0230 | 0.5130 ±0.0230 | 0.6039 ±0.0127 | 0.6804 ±0.0137 | 0.0 ±0.0 |
| DiGress+ | 10.5 ±0.6 | 97.8 ±0.8 | 8.4 ±1.1 | 7.8 ±1.2 | 0.3194 ±0.0034 | 0.3308 ±0.0041 | 0.3598 ±0.0096 | 0.4878 ±0.0155 | 0.6234 ±0.0305 | 0.6887 ±0.0250 | 6.6 ±0.9 |
| ConStruct | 6.4 ±0.6 | 99.8 ±0.2 | 100.0 ±0.0 | 99.8 ±0.2 | 0.3378 ±0.0048 | 0.3437 ±0.0104 | 0.3799 ±0.0112 | 0.5306 ±0.0150 | 0.6360 ±0.0177 | 0.6798 ±0.0436 | **88.0** ±0.5 |

general, this property is not ensured by one-shot models and cannot be included as a constraining property since it is not edge-deletion invariant.

## 4.2 Digital Pathology Cell Graphs

**Setup** In the next set of experiments, we explore digital pathology data. Due to the their natural representation of relational data, graphs are widely used to capture spatial biological dependencies from tissue images. We focus on cell graphs, whose nodes represent biological cells and edges serve as proxies for local cell-cell interactions. We build these structures from the genomic and clinical data available from the Molecular Taxonomy of Breast Cancer International Consortium (METABRIC) molecular dataset [13, 66, 15]. Each node is attributed with one of the nine possible phenotypes, which extensively characterizes a cell both anatomically and physiologically (more details in Appendix F.2). Regarding edges, we followed the typical procedure for cell graphs in digital pathology [36, 35, 2, 82]: first we employ Delaunay triangulation on the cell positions to construct the graphs, followed by edge thresholding to discard long edges. Our focus lies on generating biologically meaningful Tertiary Lymphoid Structures (TLSs), further described in Appendix F.3. Thus, we extract non-overlapping 4-hop subgraphs centered at nodes whose class is "B" from the whole-slide graphs. In terms of dimensionality, we obtain graphs with $b = 9$, corresponding to the 9 phenotypes detailed in Appendix F.2, and $c = 1$. We explore two datasets: one comprising graphs with high TLS content and another consisting of graphs with low TLS content, based on domain-specific metrics (see below). We provide their statistics in Appendix F.4. We open-source both of them, representing to the best of our knowledge the first open-source digital pathology datasets specifically tailored for graph generation. For the sake of comparison, besides ConStruct and DiGress+, we implement a non-deep learning baseline method proposed in [52] for this setting, which essentially captures 1-hop dependencies of cell graphs (see Appendix F.5). Additionally, we run BiGG [14], GraphGen [24], and SPECTRE [54]. These are the methods that, besides DiGress, can handle attributed graphs and attain non-zero V.U.N. for the planar dataset in Table 1, which we consider a proxy for performance in the digital pathology datasets due to the structural similarities (i.e., planarity) between the datasets.

**Metrics** The *TLS embedding*, $\kappa = [\kappa_0, \ldots, \kappa_5] \in \mathbb{R}^6$, has been proposed to quantify the TLS content in a cell graph [69, 52]. See Appendix F.3 for more details. Based on this metric, we define a graph $G$ to contain low TLS content if $k_1(G) < 0.05$ and high TLS content if $k_2(G) > 0.05$ [52]. To evaluate the generative performance, we adopt the average ratio for structural graph statistics (Ratio) and V.U.N. metrics used in Section 4.1. Here we consider a planar and connected graph as a valid graph. Thus, we explicitly present the proportion of generated graphs that are connected (Conn.) and (Planar). Furthermore, for a biologically meaningful evaluation of the generated cell graphs, we use the domain metrics. We report the MMD between the distributions of the components of $\kappa$. We also consider the proportion of graphs that are planar, connected, and verify the low or high TLS content condition (TLS Valid), depending on the train set used.

**Constraining criterion** We use graph planarity as target structural property for ConStruct, as cell graphs are extracted from tissue slides using Delaunay triangulation, thus necessarily planar.

**Results**   ConStruct outperforms all baselines across all summary evaluation metrics (shown in light gray in Table 2) for cell graph generation on both datasets. Unlike the synthetic datasets, here the structural distribution is conditioned on the node types, which is inherently a more complex task. This complexity contributes to the poor performance of the several unconstrained models. Constraining the edge generation process allows to significantly alleviate this modelling complexity, highlighting the benefits of ConStruct in such scenarios. We emphasize the substantial improvement in the V.U.N. of the generated graphs, with values approaching 100% using our framework, which aligns with the main motivation behind the proposed method. Interestingly, it also promotes the generation of more connected graphs. Finally, the 1-hop baseline model, while capturing the node type dependencies to some extent, as illustrated by the MMD on the components of $\kappa$, completely fails to capture structure-based dependencies.

Additionally, we carry out some experiments for molecular datasets in Appendix G: we explore the utilization of planarity for constrained molecular generation and showcase how ConStruct can be used for controlled generation. Finally, we explore likelihood-based variants of ConStruct, as well as some ablations to the projector in Appendix H.

## 5   Limitations and Future Directions

In our work, we cover edge-deletion invariant properties. However, ConStruct can be easily extended to also handle edge-insertion invariant properties (i.e., properties that hold upon edge insertion). This extension can be useful in domains where constraints such as having at least $n$ cycles in a graph are important. To achieve this, we can simply "invert" the proposed framework: design the transition matrices with the absorbing state in an existing edge state (instead of the no-edge state) and a projector that removes edges progressively (instead of inserting them) while conserving the desired property.

In the particular context of molecular generation, Appendix G illustrates that, while purely structural constraints can guide the generation of molecules with specific structural properties (e.g., acyclicity), for general properties shared by all molecules (e.g., planarity) they are too loose. In contrast, autoregressive models thrive in such setting due to the possibility of molecular node ordering (e.g., via canonical SMILES) and the efficient incorporation of *joint node-edge* constraints (e.g., valency). Therefore, although it consists of a fundamentally different setting than the one considered in this paper, incorporating joint node-edge constraints into ConStruct represents an exciting future direction.

Additionally, the induced sparsity created by the edge-absorbing noise model presents opportunities for further exploitation. By leveraging this sparsity, future extensions of ConStruct could enhance sampling efficiency and improve the underlying diffusion model's scalability for generating larger graphs.

## 6   Conclusion

In this paper, we introduced ConStruct, a framework that allows to integrate domain knowledge via structural constraints into graph diffusion models. By constraining the diffusion process based on a diverse set of geometric properties, we enable the generation of realistic graphs in scenarios with limited data. To accomplish that, we leverage an edge-absorbing noise model and a projector operator to ensure that both the forward and reverse processes preserve the sampled graphs within the constrained domain and, thus, maintain their validity. Despite its algorithmic simplicity, our approach overcomes the arbitrarily hard problem of projecting a given graph into a combinatorial subspace in an efficient and theoretically grounded manner. Through several experiments on benchmark datasets, we showcase the versatility of ConStruct across various structural constraints. For example, in digital pathology datasets, our method outperforms existing approaches, bringing the validity of the generated graphs close to 100%. Overall, ConStruct opens new avenues for integrating domain-specific knowledge into graph generative models, thereby paving the way for their application in real-world scenarios.

## Acknowledgements

We thank Cédric Vincent-Cuaz, Nikolaos Dimitriadis, Vaishnavi Subramanian, Yiming Qin, Sevda Öğüt, and Laura Toni for helpful discussions and feedback. We also thank Andreas Bergmeister and Yunhui Jang for helping set up the code to reproduce their experiments.

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

# A   Graph Discrete Diffusion Model

In this section, we further detail the design of the graph discrete diffusion model used to illustrate the constraining framework of ConStruct.

## A.1   Training Algorithm

The denoising neural network is trained using the cross-entropy loss between its predicted probabilities for each node and edge types, $\hat{p}^G = (\hat{p}^X, \hat{p}^E)$ and the actual node and edge types of a clean graph, $\mathbf{G} = (\mathbf{X}, \mathbf{E})$:

$$L(\hat{p}^G, G) = \mathrm{CE}\left(\hat{p}^X, \mathbf{X}\right) + \lambda \, \mathrm{CE}\left(\hat{p}^E, \mathbf{E}\right), \tag{4}$$

where $\lambda$ is an hyperparameter that is tuned to balance both loss terms.

As shown by Vignac et al. [79], the loss in Equation (4) is node permutation invariant. Thus, if we also consider an equivariant architecture, the diffusion model is endowed of the desired equivariance properties, allowing the model to dodge the node ordering sensitivity from which, for example, autoregressive models suffer. For this reason, we adopt a Graph Neural Network, $\mathrm{GNN}_\theta$, as the denoising neural network of our diffusion model. In particular, we employ the exact same denoising network architecture of DiGress [79], a Graph Transformer [20].

Importantly, the edge-absorbing noise model used in ConStruct increases graph sparsity throughout the forward trajectory. Consequently, beyond the distribution preserving guarantees, it also allows for the efficient computation of extra features on the noisy graphs that otherwise the $\mathrm{GNN}_\theta$ would not be able to capture. Following Vignac et al. [79], these are fed as a supplementary input to the denoising network (see Algorithm 1 and Algorithm 3), further enhancing its expressivity beyond the well-known limited representational power of GNN architectures [84, 57]. More concretely, besides the spectral (eigenvalues and eigenvectors of the Laplacian) and structural (number of cycles) features from DiGress, we also consider some additional features. We add as graph features the degree distribution and the node and edge type distributions. While the former enhances the positional information within the graph, the latter helps in making more explicit to the model the prevalence of each class in the dataset. Additionally, we add auxiliary structural encodings to edges to boost edge label prediction. We compute the Adamic-Adar index to aggregate local neighborhood information and the shortest distance between nodes to encode node interactions. Due to computational limitations, we only consider information within a 10-hop radius for these computations. These additional features were previously proposed by Qin et al. [64].

Provided such loss function and denoising neural network architecture, all the necessary elements are in place for the training of the diffusion model, which is defined in Algorithm 1.

---

**Algorithm 1:** Training Algorithm for Graph Discrete Diffusion Model

**Input:** Graph dataset $\mathcal{D}$

1  **repeat**
2       Sample $G = (\mathbf{X}, \mathbf{E}) \sim \mathcal{D}$;
3       Sample $t \sim \mathcal{U}(1, ..., T)$;
4       Sample $G^t \sim \mathbf{X}\bar{\mathbf{Q}}_X^t \times \mathbf{E}\bar{\mathbf{Q}}_E^t$;
5       $h \leftarrow f(G^t, t)$ ;                   `// Compute extra features`
6       $\hat{p}^X, \hat{p}^E \leftarrow \mathrm{GNN}_\theta(G^t, h)$;
7       $\mathrm{loss} \leftarrow \mathrm{CE}(\hat{p}^X, \mathbf{X}) + \lambda \, \mathrm{CE}(\hat{p}^E, \mathbf{E})$;
8       $\mathrm{optimizer.step(loss)}$;
9  **until** convergence of $\mathrm{GNN}_\theta$;

---

## A.2   Parameterization of the Reverse Process

The distribution $p_\theta(G^{t-1}|G^t)$ fully defines the reverse process. Under an independence assumption between nodes and edges, this distribution can be modelled as:

$$p_\theta(G^{t-1}|G^t) = \prod_{1 \le i \le n} p_\theta(x_i^{t-1}|G^t) \prod_{1 \le i,j \le n} p_\theta(e_{ij}^{t-1}|G^t). \tag{5}$$

To compute each of these terms, we use the GNN$_\theta$ predictions through the following marginalization:

$$p_\theta(x_i^{t-1}|G^t) = \sum_{x \in \mathcal{X}} p_\theta(x_i^{t-1}|x_i = x, G^t)\ \hat{p}_i^X(x), \tag{6}$$

where $\hat{p}_i^X(x)$ denotes the GNN$_\theta$ predicted probability of node $i$ being of type $x$. Similarly, for the edges we have $p_\theta(e_{ij}^{t-1}|G^t) = \sum_{e \in \mathcal{E}} p_\theta(e_{ij}^{t-1}|e_{ij} = e, G^t)\ \hat{p}_{ij}^E(e)$. To compute the missing term in Equation (6), we equate it to the posterior term of the forward process:

$$p_\theta(x_i^{t-1}|x_i = x, G^t) = \begin{cases} \frac{\mathbf{x}_i^t(\mathbf{Q}_X^t)' \odot \mathbf{x}_i \bar{\mathbf{Q}}_X^{t-1}}{\mathbf{x}_i^t \bar{\mathbf{Q}}_X^t \mathbf{x}_i} & \text{if } q(x_i^t|x_i = x) > 0, \\ 0 & \text{otherwise,} \end{cases} \tag{7}$$

where $'$ denotes transposition.

### A.3 Sampling Algorithm

In this section, we first introduce the algorithm describing the proposed projector operator in Algorithm 2. This projector is employed at each time step to keep the sampled graphs throughout the reverse process within the constrained domain. The full sampling algorithm is shown in Algorithm 3.

---

**Algorithm 2:** Projector

**Input:** Constraining property $P$, noisy graph $G^t = (X^t, E^t)$, and candidate graph
$\hat{G}^{t-1} = (\hat{X}^{t-1}, \hat{E}^{t-1})$

1   $G^{t-1} \leftarrow (\hat{X}^{t-1}, E^t)$;
2   $E' \leftarrow \hat{E}^{t-1} \setminus E^t$ ;                      `// Get candidate edges`
3   **repeat**
4      Sample $e' \sim E'$;
5      **if** $P(G^{t-1}.\text{insert}(e'))$ **then**
6         $G^{t-1} \leftarrow G^{t-1}.\text{insert}(e')$ ;          `// Insert only valid edges`
7      **end**
8      $E' \leftarrow E' \setminus \{e'\}$;
9   **until** $E' = \emptyset$;
10   **return** $G^{t-1}$;

---

**Algorithm 3:** Sampling Algorithm for Constrained Graph Discrete Diffusion Model

**Input:** Number of graphs to sample $N$ and constraining property $P$

1   **for** $i = 1$ **to** $N$ **do**
2      Sample $n$ from the training set distribution ;         `// Sample number of nodes`
3      Sample $G^T \sim q_X(n) \times q_E(n)$ ;        `// Sample from limit distribution`
4      **for** $t = T$ **to** $1$ **do**
5         $h \leftarrow f(G^t, t)$ ;                `// Compute extra features`
6         $\hat{p}^X, \hat{p}^E \leftarrow \text{GNN}_\theta(G^t, h)$;
7         $\hat{G}^{t-1} \sim p_\theta(G^{t-1}|G^t)$ ;     `// Sample from distribution in Equation (5)`
8         $G^{t-1} \leftarrow \text{Projector}(P, \hat{G}^{t-1}, G^t)$;
9      **end**
10      Store $G^0$;
11   **end**

---

# B  Theoretical Analysis

In this section, we theoretically analyse the projector. We start by defining a notion of distance between graphs [68].

**Definition B.1.** *Let $G_1$ and $G_2$ be two unattributed graphs. The graph edit distance with uniform cost, denoted by $\mathrm{GED}(G_1, G_2)$, is defined as:*

$$\mathrm{GED}\,(G_1, G_2) = \min_{(e_1,\ldots,e_k) \in \mathcal{E}(G_1,G_2)} \sum_{i=1}^{k} c\,(e_i) = \min_{(e_1,\ldots,e_k) \in \mathcal{E}(G_1,G_2)} \alpha|(e_1,\ldots,e_k)| \qquad (8)$$

*where $\mathcal{E}(g_1, g_2)$ denotes the set of edit paths that convert $G_1$ into $G_2$ (up to an isomorphism), $c(e) = \alpha > 0$ is the uniform cost of each usual set of elementary graph edit operators and $|(e_1,\ldots,e_k)|$ refers to the cardinality of the edit path.*

Importantly, in this analysis we choose GED due to its permutation invariance properties. We only define it over unattributed graphs for an objective evaluation as ConStruct only operates at the graph structural level. Moreover, as our generative process imposes a fixed number of nodes throughout the whole reverse process, the relevant elementary edits for GED are edge insertion and deletion.

Additionally, we use the notation $G \supset G'$ to denote that $G' = (X', E')$ is a subgraph of $G = (X, E)$, i.e., that up to an isomorphism, we have $E \supset E'$ and $X = X'$. For brevity, we slightly abuse notation and also define the union between a graph, $G = (X, E)$, and a set of edges, $E'$, to be the graph whose edges result from the union of its edges with those of the set, i.e., $G \cup E' = G' = (X, E \cup E')$. Similarly, we have $G \setminus E' = G' = (X, E \setminus E')$.

The next results are organized in the following way: the first theorem proves that for any edge-deletion invariant constraining property (Definition 3.1), our projector can retrieve a graph that results from a projection onto the constrained set under the GED sense. Then, we prove that when considering acyclicity as target structural property, the projector is guaranteed to output the optimal (projected) samples. We finally show that this second property does not hold for all edge-deletion invariant properties, giving counter-examples for the cases of planarity, maximum degree and lobster components.

**Theorem 1.** *Let:*

- *$P$ be the edge-deletion invariant (Definition 3.1) constraining property of the projector;*
- *$G^t$ be a noisy graph obtained at timestep $t$;*
- *$\hat{G}^{t-1}$ be a sampled graph from $p_\theta(G^{t-1}|G^t)$, i.e., the one-step denoised candidate graph directly proposed by the diffusion model when taking $G^t$ as input;*
- *$\mathcal{G}^{t-1} = \mathrm{Projector}(P, \hat{G}^{t-1}, G^t)$ be the set of all possible final one-step denoised graph outputted by ConStruct.*

*We define $(G^*, e^*)$ any optimal solutions of the following optimization problem:*

$$\min_{G \in \mathcal{C}} \mathrm{GED}(\hat{G}^{t-1}, G) = \min_{G \in \mathcal{C}} \min_{(e_1,\ldots,e_k) \in \mathcal{E}(\hat{G}^{t-1},G)} \alpha|(e_1,\ldots,e_k)|, \qquad (9)$$

*where $\mathcal{C} = \{G \in \mathcal{G} | P(G) = True, G \supset G^t\}$, with $\mathcal{G}$ the set of all unattributed graphs. Then, $(G^*, e^*)$ can be recovered by our projector, i.e. $G^* \in \mathcal{G}^{t-1}$.*

*Proof.* If $\hat{G}^{t-1} \in \mathcal{C}$, the theorem is trivially verified since the output of the projector is directly $\hat{G}^{t-1}$, as well as the solution of the minimization problem. Therefore, for the rest of the proof, we only consider the case $\hat{G}^{t-1} \notin \mathcal{C}$.

Now, since the reverse process of the diffusion model is an edge insertion process, we have $\hat{G}^{t-1} = G^t \cup E_{\mathrm{candidate}} \supset G^t$. Also, we notice that the projector amounts to randomly remove the edges that are not in $G^t$ from $\hat{G}^{t-1}$ until we find a graph within the constraint set (equivalently, it entails adding as many edges as possible to $G^t$ while ensuring that the graph remains within the constraint set). Thus, it suffices to prove that for $(G^*, e^*)$ we necessarily have an optimal edit path $e^* = (e_1^*, \ldots, e_k^*)$ from $\hat{G}^{t-1}$ to $G^*$ exclusively composed of edge deletions. In this case, our projector can necessarily produce $G^*$.

Let $(G^*, e^*)$ be such a solution, and define $G^*_{:i}$ the graph resulting from the $i^{th}$ first edits $e^*_{:i} = (e^*_1, ..., e^*_i)$ with $i \leq k$. We will prove by induction that, for all $1 \leq i \leq k$, $e^*_{:i}$ is only composed of edge deletions such that $G^t \subset G^*_{:i}$.

$\mathbf{i = 1}$: Since $\hat{G}^{t-1} \notin \mathcal{C}$ and $\hat{G}^{t-1} \supset G^t$, we have that $P(\hat{G}^{t-1}) = False$. As $P$ is edge-deletion invariant, inserting any set of edges $E$ to $\hat{G}^{t-1}$ implies that

$$P(\hat{G}^{t-1}) = False \implies P(\hat{G}^{t-1} \cup E) = False. \tag{10}$$

Therefore, we have:

$$\min_{G \in \mathcal{C}} \text{GED}(\hat{G}^{t-1}, G) = \min_{(G^t \cup E_G) \in \mathcal{C}} \text{GED}(G^t \cup E_{\text{candidate}}, G^t \cup E_G) \tag{11}$$

$$= \min_{(G^t \cup E_G) \in \mathcal{C}} \alpha |E_{\text{candidate}} \setminus E_G| \tag{12}$$

$$\leq \min_{(G^t \cup E_G) \in \mathcal{C}} \alpha |E_{\text{candidate}} \cup E \setminus E_G| \tag{13}$$

$$= \min_{(G^t \cup E_G) \in \mathcal{C}} \text{GED}(\hat{G}^{t-1} \cup E, G^t \cup E_G) \tag{14}$$

$$= \min_{G \in \mathcal{C}} \text{GED}(\hat{G}^{t-1} \cup E, G). \tag{15}$$

Thus, we conclude that any edge insertions would take us further away from the constraint set. Therefore, $e^*_1$ cannot represent an edge insertion. However, it could still be an edge deletion such that $G^t \not\subset G^*_{:1}$ In this case, an extra edge insertion would be necessary to recover $G^t$ in $G^*_{:1}$, which is required since $G^* \supset G^t$, i.e.,

$$\min_{G \in \mathcal{C}} \text{GED}(\hat{G}^{t-1}, G) \leq \min_{G \in \mathcal{C}} \text{GED}(G^*_{:1}, G). \tag{16}$$

Contrarily, if $e^*_1$ is an edge deletion such that $G^t \subset G^*_{:1}$, we have:

$$\min_{G \in \mathcal{C}} \text{GED}(\hat{G}^{t-1}, G) > \min_{G \in \mathcal{C}} \text{GED}(G^*_{:1}, G), \tag{17}$$

since $G^t \subset G^*_{:1} \subset \hat{G}^{t-1}$. Therefore, we verify the intended property for $i = 1$.

$\mathbf{1 < i \leq k}$: We have $G^*_{:i-1} \notin \mathcal{C}$ because $P(G^*_{:i-1}) = False$. Otherwise $G^*_{:i-1}$ would be the solution since $G^t \subset G^*_{:i-1}$. Hence for any set of inserted edges $E$,

$$\min_{G \in \mathcal{C}} \text{GED}(G^*_{:i-1}, G) \leq \min_{G \in \mathcal{C}} \text{GED}(G^*_{:i} \cup E, G), \tag{18}$$

implying that $e^*_i$ is an edge deletion. By the same token, if $G^t \not\subset G^*_{:i}$, then necessarily an extra insertion edit would be necessary to recover $G^t$ further in $G^*$, so we have again:

$$\min_{G \in \mathcal{C}} \text{GED}(G^*_{:i-1}, G) \leq \min_{G \in \mathcal{C}} \text{GED}(G^*_{:i}, G), \tag{19}$$

which, as seen before, is suboptimal. Thus, $e^*_i$ is an edge deletion such that $G^t \subset G^*_{:i}$. By noticing that $G^*_{:k} = G^*$, we conclude our proof. This induction shows that only an edit path $e^*$ composed of edge deletions such that all intermediate graphs contain $G^t$ leads to an optimal projection w.r.t GED. $\square$

**Critical analysis of the result in Theorem 1**: See Section 3.4 for a critical analysis of this result.

In the following theorem we prove that the projector always picks the solution of the optimization problem, i.e., that any element of $\mathcal{G}^{t-1}$ is actually a solution of the optimization problem in Theorem 1. In this proof, we use the concept of connected component of a graph, i.e., a subgraph of the given graph in which there is a path between any of its two vertices, but no path exists between any vertex in the subgraph and any vertex outside of it. Therefore, any edge inserted between two nodes in the same connected component leads to a cycle. Importantly, we trivially consider an isolated node as a connected component.

**Theorem 2.** *Under the same conditions of Theorem 1, if $P$ returns true for graphs with no cycles, we have:*

$$\mathcal{G}^{t-1} = \text{argmin}_{G \in \mathcal{C}} \text{GED}(\hat{G}^{t-1}, G).$$

*Proof.* Following the proof of Theorem 1, if we define again $\hat{G}^{t-1} = G^t \cup E_{\text{candidate}}$, we have:

$$\min_{G \in \mathcal{C}} \text{GED}(\hat{G}^{t-1}, G) = \min_{(G^t \cup E_G) \in \mathcal{C}} \text{GED}(G^t \cup E_{\text{candidate}}, G^t \cup E_G)$$

$$= \min_{\{E_G \mid P(G^t \cup E_G) = \text{True}, \ E_G \subset E_{\text{candidate}}\}} |E_{\text{candidate}} \setminus E_G|$$

$$= \max_{\{E_G \mid P(G^t \cup E_G) = \text{True}, \ E_G \subset E_{\text{candidate}}\}} |E_G|,$$

where the first equality is just a change of variables and the second comes from $E_{\text{candidate}} \supset E_G$, as shown in Theorem 1. This shows that a solution to $\text{argmin}_{G \in \mathcal{C}} \text{GED}(\hat{G}^{t-1}, G)$ maximizes the number of edges added to $G^t$. This is a general result under the conditions of Theorem 1 and not specific for graphs with no cycles. We now want to show that any element of $\mathcal{G}^{t-1}$ is a solution to this optimization problem.

We define $|\text{CC}_{\text{candidate}}|$ as the number of distinct connected components of $G^t$ reached by $E_{\text{candidate}}$. We remark that the considered graphs all have the same fixed number of nodes. A well-known result for graphs without cycles is that, under the provided setting, the maximum number of edges from $E_{\text{candidate}}$ that we can insert is $|\text{CC}_{\text{candidate}}| - 1$, i.e., we sequentially insert an edge per pair of separate connected components. Thus, we have:

$$\max_{\{E_G \mid P(G^t \cup E_G) = \text{True}, \ E_G \subset E_{\text{candidate}}\}} |E_G| \leq |\text{CC}_{\text{candidate}}| - 1.$$

On the other hand, the only edges from $E_{\text{candidate}}$ that the projector rejects are the ones creating cycles. This means that the refused edges would not reduce the number of separate connected components, since they would connect vertices already in the same connected component. Thus, for a graph $G_{\text{projector}} = G^t \cup E_{\text{projector}} \in \mathcal{G}^{t-1}$, we necessarily have:

$$|E_{\text{projector}}| \geq |\text{CC}_{\text{candidate}}| - 1,$$

which tightly matches the upper bound for the optimization problem seen above. Consequently, $|E_{\text{projector}}| = |\text{CC}_{\text{candidate}}| - 1$ and any $G_{\text{projector}} \in \mathcal{G}^{t-1}$ is necessarily solution of the optimization problem, i.e.:

$$\mathcal{G}^{t-1} \subset \text{argmin}_{G \in \mathcal{C}} \text{GED}(\hat{G}^{t-1}, G)$$

$\square$

For other edge-deletion invariant properties, we provide examples of $G_{\text{projector}}$ with a different number of edges inserted by the projector, $|E_{\text{projector}}|$, in Figure 3. These are necessarily counter-examples to what was proved in Theorem 2 for acyclic graphs ($\mathcal{G}^{t-1} \subset \text{argmin}_{G \in \mathcal{C}} \text{GED}(\hat{G}^{t-1}, G)$). Nevertheless, the opposite relation still holds from Theorem 1.

Overall, Theorem 1, Theorem 2, and the counter-examples in Figure 3 show that the problem that the projector is addressing is not so trivial such that it can always output the optimal graph in the GED sense for all the edge-deletion invariant properties. Nevertheless, our projector is still guaranteed to produce graphs that meet the specified structural constraints.

| Constraint | $G^t$ | $\hat{G}^{t-1}$ | $G^{t-1}$ | $\left|E_{\text{Projector}}\right|$ |
|---|---|---|---|---|
| Planar | | | | 2 |
| | | | | 1 |
| Maximum Degree | | | | 2 |
| | | | | 1 |
| Lobster Components | | | | 6 |
| | | | | 5 |

Figure 3: Examples of different $G^{t-1}$ that can be yielded by $\text{Projector}(P, \hat{G}^{t-1}, G^t)$ for given $P$ (column "Constraint"), $G^t$, and $\hat{G}^{t-1}$ (columns with the respective name) that lead to the insertion of a different number of edges. For the maximum degree row, the example given considers that the maximum allowed degree is 2. For the column $\hat{G}^{t-1}$, the dashed lines represent the candidate edges. For the column $G^{t-1}$, the green lines denote the actually inserted edges by the projector.

## C Incremental Algorithms

As discussed in Section 3.5, utilizing the projector in the edge insertion reverse process (i.e., $G^t \subset G^{t-1}$) allows us to further enhance efficiency by leveraging incremental algorithms for graph property satisfaction checking. These algorithms avoid performing a full property satisfaction check on the new graph at each timestep. Instead, they assume that the previous graph (i.e., before the new edge was added) already satisfies the target structural property. Incremental algorithms focus on verifying the impact of the newly added edge by updating and checking only the affected parts of smartly designed data structures. In other words, contrary to their full graph counterparts, incremental algorithms allow for property satisfaction checks at a local level. This approach accelerates the property satisfaction checking process by reducing redundant computation.

In this section, we discuss the incremental property satisfaction algorithms for the edge-deletion invariant properties analysed throughout the paper. We also note that due to the combinatorial nature of edge-deletion invariant properties, each property satisfaction algorithm is specific to the property in question. There is no general efficient property satisfaction checker for all edge-deletion invariant properties. Consequently, we address each property on a case-by-case basis.

**Planar** The best performing full property satisfaction algorithm known for planarity is $O(n)$ [30], while its fastest known incremental test has amortized running time of $O(\alpha(\mathfrak{q}, n))$ [43] ("almost constant" complexity), where $\mathfrak{q}$ is the total number of operations (edge queries and insertions), and $\alpha$ denotes the inverse-Ackermann function.

**Acyclicity** In generic undirected graphs (our case), the usual full tests via DFS/BFS have a complexity of $O(n + |E|)$, i.e., the algorithms have to traverse the full graph to reject the existence of any cycle. However, for the dynamic case, given that $G^t$ has no cycles, we can only check if the added edges, $E_{\text{added}}$, connect nodes already in the same connected component. This check can be efficiently performed if we keep an updated hashtable that maps each node to the index of the connected component it belongs at that iteration (an isolated node is a connected component) and another one with all the nodes belonging to each connected component. Whenever there is a new edge proposed, we check if the nodes are already in the same connected component. If not, we insert the edge and update the two hashtables accordingly; otherwise, we reject the edge since it would create a cycle. Therefore, the cycle check can be done in $O(|E_{\text{added}}|)$.

**Lobster Components** Its global test involves removing twice the leaves of the graph and checking if the remaining connected components are paths. This algorithm has a complexity of $O(|E|)$. For the incremental version, we can use a similar approach to that of the absence of cycles but additionally check if the newly connected node is not more than two hops away from the path in its connected component, as lobster graphs are specific instances of forest graphs. If, again, we keep track of the paths of each connected component in a hashtable, we still get an incremental algorithm of complexity $O(|E_{\text{added}}|)$ for this property.

**Maximum Degree** The optimal full property satisfaction algorithm has a complexity of $O(n)$ since it has to perform a degree check across all nodes. The incremental version is naturally just a quick check for nodes that are vertices of $E_{\text{added}}$. Again, if we keep an updated hashtable with the degree of each node, this can be quickly performed in $O(|E_{\text{added}}|)$.

# D    Experimental Details

## D.1    Training Details

As mentioned in Section 4, we follow the splits originally proposed for each of the unattributed datasets (lobster [46], planar [54], and tree [7]): 80% of the graphs are used in the train set and the remaining 20% are allocated to the test set. We use 20% of the train set as validation set. We note that for the lobster dataset, the original splits provided in the open-source code from Liao et al. [46] use the validation set as a subset of the train set, i.e., all the samples in the validation set are used to train. In contrast, we follow Martinkus et al. [54]'s protocol, isolating completely the validation samples from the train (again, 20% of the train split). In any case, the test splits are coincident between our approach and the one from Liao et al. [46]. For the digital pathology datasets, we follow the same protocol.

These splits and the hyperparameters used for each model are provided in the computational implementation of the paper as the default values for each of the experiments. For each configuration, we save the five best models in terms of negative log likelihood and the last one (for ConStruct, we compute the likelihood of the corresponding unconstrained model) and pick the best performing model across those six checkpoints. Regarding the optimizer, we used the AMSGrad [65] version of AdamW [49] with a learning rate of 0.0002 and weight decay of 1e-12 for all the experiments.

## D.2    Resources

All our experiments were run in a single Nvidia V100 32Gb GPUs. We present the training times of the diffusion model for each dataset in Table 3.

Table 3: Training times for the diffusion model in different datasets.

| Dataset | Training Time (h) |
| --- | --- |
| Planar | 48 |
| Tree | 44 |
| Lobster | 50 |
| High TLS | 61 |
| Low TLS | 61 |
| QM9 | 9.5 |
| MOSES | 335.5 |
| GuacaMol | 502 |

The baseline model for the digital pathology dataset does not use any GPU. It takes 0.6s to train and 2 minutes to sample from. The sampling times for ConStruct can be found in Appendix D.3. As the order of magnitude of training times is significantly larger than the one of sampling times, Table 3 provides a good estimate of the total computational resources required for this paper.

## D.3    Runtimes

A major advantage of our framework is that it does not interfere with the training of the diffusion model, preserving its efficiency. Therefore, there is no overburden in the training time caused by ConStruct. For this reason, in this section we only analyse the different sampling runtimes. In particular, we track the sampling times of DiGress+ and the ones of ConStruct with and without the efficiency boosting components described in Section 3.5 (edge blocking hashtable and incremental property satisfaction algorithm).

Additionally, a natural procedure to ensure 100% constraint verification solely using DiGress+ is to first directly perform unconstrained generation and then applying a validation process to filter out the ones that do not verify the constraint. This *a posteriori* filtering requires a full graph property check, run only once after the graph has been generated but that is thus more computationally expensive when compared to their incremental versions employed by ConStruct. The complexity comparison between a full graph property check ran only once *vs* an incremental check for each added edge is property specific. However, the main bottleneck of the *a posteriori* filtering is that it

wastes computational resources in case of graph rejection (i.e., in the case of a graph not verifying the property, all the resources used in its generation and full graph property checking are wasted) and requires restarting the sampling from zero again with an additional property check at the end. Furthermore, this procedure has to be performed sequentially, since we can only check the graphs constraint satisfaction after their generation. ConStruct avoids such redundancy and, thus, waste of computation by generating property satisfying graphs by design: throughout the reverse process, we know that the previous graph verified the property, so we can just check property satisfaction for the newly added edges (via incremental algorithm) that have not been checked yet (via edge blocking hashtable).

We compare the sampling runtimes of the aforementioned algorithmic variants in the table below. We run this experiment for the tree dataset, where we use acylicity as target structural property. We picked this dataset due to its simpler incremental check, described in Appendix C.

Table 4: Runtimes comparison. We performed five sampling runs for each method and present their results in the format mean ± standard error of the mean. For each run, we generated 100 graphs. All our experiments were run in a single Nvidia V100 32Gb GPUs. ConStruct [efficient] uses the edge blocking hashtable and the incremental version of the property satisfaction algorithm, while ConStruct [baseline] does not. DiGress+ refers to regular unconstrained generation, while DiGress+ [rejection] applies *a posteriori* filtering of unconstrained generation until we get the intended amount of graphs.

| Dataset | Sampling Time (s) |
|---|---|
| DiGress+ | $266.0_{\pm 0.1}$ |
| DiGress+ [rejection] | $310.7_{\pm 5.5}$ |
| ConStruct [efficient] | $290.2_{\pm 0.1}$ |
| ConStruct [baseline] | $349.0_{\pm 0.3}$ |

By implementing the edge blocking hashtable and the incremental checker, we observe a significant efficiency improvement: the additional runtime imposed by ConStruct over the unconstrained setting decreases from 31% to 9%. This trend should hold for other datasets as far as both lookup and update operations in hashtables ($O(1)$) and incremental property checks are more efficient than full-graph constraint checks, which is the typical case.

ConStruct also outperforms the *a posteriori* filtering of unconstrained generation. In this case, we used an unconstrained model that generates constraint satisfying properties 97% (DiGress+ in Table 1 of the paper), therefore largely benefitting the unconstrained model. For example, if we considered the digital pathology setting, where we can have only 6.6% (see Table 2, high TLS dataset, DiGress+) of the generated graphs with the unconstrained model satisfying the constraint, the amount of wasted computation would be dramatically larger, implying a much worse runtime. In such setting, ConStruct would be approximately 12 times more efficient in generating valid graphs than DiGress+. Additionally, this gap in sampling efficiency may become particularly critical in settings where the amount of generated graphs is much larger, as is the case for molecular generation (two orders of magnitude greater than in the settings with synthetic datasets, see Appendix G).

# E  Synthetic Datasets

In this section, we provide further information about the unattributed synthetic datasets used in Section 4.1.

## E.1  Statistics

In Table 5, we provide the minimum, maximum, and average number of nodes, minimum, maximum, and average number of edges, and the number of training, validation and test graphs used for each synthetic unattributed dataset.

Table 5: Synthetic dataset statistics. #Train, #Val and #Test denote the number of graphs considered in the train, validation and test splits, respectively.

| Dataset | Min. nodes | Max. nodes | Avg. nodes | Min. edges | Max. edges | Avg. edges | #Train | #Val | #Test |
|---------|-----------|-----------|-----------|-----------|-----------|-----------|--------|------|-------|
| Planar  | 64 | 64 | 64   | 173 | 181 | 177.8 | 128 | 32 | 40 |
| Tree    | 64 | 64 | 64   | 63  | 63  | 63    | 128 | 32 | 40 |
| Lobster | 11 | 99 | 50.2 | 10  | 99  | 49.2  | 64  | 16 | 20 |

## E.2  Compared Methods

In Section 4, we compare ConStruct with several unconstrained graph generative models. We consider:

- the two first widely adopted autoregressive models for graph generation, GraphRNN [87] and GRAN [46];
- two spectrally conditioned methods: SPECTRE [54] is a GAN-based approach and HSpectre [7] consists of an iterative local expansion method that takes advantage of a score-based formulation for intermediate steps;
- we also compare to the original implementation of DiGress [79] without the additional features described in Appendix A.1;
- GraphGen [24] is a scalable autoregressive method based on graph canonization through minimum DFS codes. Importantly, this method is domain-agnostic and supports attributed graphs by default;
- GraphGen-Redux [5] improves over GraphGen by jointly modelling the node and edge labels;
- BwR [17] and GEEL [34] also explore more scalable graph representations via bandwidth restriction schemes, which are then fed to other graph generation architectures;
- HDDT [33] leverages a $K^2-$tree representation of graphs to capture their hierarchical structure in an autoregressive manner;
- GDSS [38] is a purely score-based formulation for graph generation;
- BiGG [14] is a parallelizable autoregressive model that takes advantage of graph sparsity to scale for large graphs;
- EDGE [10] is a degree-guided scalable discrete diffusion method (more details in Section 2).

# F   Digital Pathology

In this section, we go through additional information related to the digital pathology datasets.

## F.1   Digital Pathology Primer

Digital pathology consists of an advanced form of pathology that involves digitizing tissue slides into whole-slide images (WSI), allowing for computer-based analysis and storage. Deep learning approaches quickly integrated digital pathology processing methods, primarily focusing on extracting image-level representations for tasks such as slide segmentation and structure detection. These have also been used for downstream tasks such as cancer grading, or survival prediction [6, 71]. However, existing image-based approaches face challenges with the sizes of WSIs, requiring their patching. This procedure raises a trade-off between the context and the size of the patch provided to the model. Moreover, image-based deep learning lacks efficient representations of biological entities and their relations, resulting in less interpretable models. Recently, entity-graph based approaches have emerged as a promising alternative to evade such limitations [36, 2]. These graphs are built by directly assigning nodes to biological entities and modelling their interactions with edges [26, 36], providing enhanced predictive performance and interpretability [35, 82].

Importantly, most of the deep learning contributions in the digital pathology realm have been in the discriminative setting. However, digital pathology could profoundly benefit from the development of generative formulations in several dimensions: first, there is is a lack of high-quality annotated samples, mostly due to their heavy ethical and privacy regulation. Besides, collecting these samples is remarkably costly, both economically and in terms of time and labor required [36]. Most of the discriminative approaches are also instance-based. The developed models then become highly sensitive to distribution shifts, which is a common challenge across biomedical datasets, for instance due to batch effects [21]. The development of generative models in digital pathology can address these limitations by enabling both the generation of synthetic data and distribution-based characterisations of the data. Even though some approaches have been carried out using image-based methods (e.g., GANs [39] or even diffusion models [56]), these lack the advantages of graph-based approaches. To the best of our knowledge, graph-based generative modelling in digital pathology has only been explored by Madeira et al. [52]. Despite the promising results for data augmentation settings, only an off-the-shelf graph generative model (DiGress) is explored and in a proprietary dataset.

## F.2   Building Whole-slide Cell Graphs

We build the whole-slide cell graphs from the genomic and clinical data available from the Molecular Taxonomy of Breast Cancer International Consortium (METABRIC) molecular dataset[3]. This dataset has been extensively used in previous breast cancer studies [66, 13, 15]. Using the single cell data, we mapped 32 different annotated cell phenotypes to 9 more generic phenotypes in a biologically grounded manner. We used the mapping detailed in Table 6. Therefore, each node is assigned to one of the resulting nine possible phenotypes. We assume these phenotypes to extensively characterize a cell both anatomically and physiologically.

Regarding edges, we followed the typical procedure for cell-graphs in digital pathology [36, 35, 2, 82]: first we used Delaunay triangulation on the cell positions to build them. Then, we discard edges longer than 25 $\mu m$. We note that we obtain different graphs than the ones considered by Danenberg et al. [15]. In terms of dimensionality, we obtain graphs with $b = 9$ and $c = 1$. We focus on the generation of simple yet biologically meaningful structures, Tertiary Lymphoid Structures (TLSs), further described in the next section. Thus, we extract 4-hop non-overlapping subgraphs centered at nodes whose class is "B-cell" from the whole-slide graphs.

## F.3   Tertiary Lymphoid Structures

Tertiary Lymphoid Structures (TLSs) are simple yet biologically meaningful structures. Structurally, TLSs are well-organized biological entities where clusters of B-cells are enveloped by supporting T-cells. Typically observed in ectopic locations associated with chronic inflammation [61, 69], these structures have been linked to extended disease-free survival in cancer [28, 44, 18, 58, 69], thus

---

[3]Data retrieved from https://zenodo.org/records/7324285

Table 6: Mapping used to convert the original phenotypes to the adopted phenotypes.

| Original Phenotype | Mapping Phenotype |
|---|---|
| CK8-18$^{hi}$CXCL12$^{hi}$ | Epithelial |
| HER2$^+$ | |
| MHC$^{hi}$CD15$^+$ | |
| CK8-18$^{hi}$ER$^{lo}$ | |
| CK$^{lo}$ER$^{lo}$ | |
| CK$^{lo}$ER$^{med}$ | |
| CK8-18$^+$ ER$^{hi}$ | |
| CK$^{med}$ER$^{lo}$ | |
| MHC I & II$^{hi}$ | |
| Basal | |
| Ep CD57$^+$ | |
| MHC I$^{hi}$CD57$^+$ | |
| ER$^{hi}$CXCL12$^+$ | |
| Ep Ki67$^+$ | |
| CK$^+$ CXCL12$^+$ | |
| CD15$^+$ | |
| Endothelial | Endothelial |
| Macrophages & granulocytes | Macrophages/Granulocytes |
| Macrophages | |
| Granulocytes | |
| Fibroblasts | Fibroblast |
| Fibroblasts FSP1$^+$ | |
| Myofibroblasts | Myofibroblast |
| Myofibroblasts PDPN$^+$ | |
| CD4$^+$ T cells | T |
| CD4$^+$ T cells & APCs | |
| CD8$^+$ T cells | |
| T$_{Reg}$ & T$_{Ex}$ | |
| B cells | B |
| CD57$^+$ | Marker |
| Ki67$^+$ | |
| CD38$^+$ lymphocytes | CD38+ Lymphocyte |

constituting an important indicator for medical prognosis in cancer. Since these are small structures when compared with the size of whole-slide graph, we extract non-overlapping 4-hop subgraphs centered at nodes whose class is "B", corresponding to B cells, from the WSI graphs. This procedure is illustrated in Figure 4.

As mentioned in Section 4.2, the TLS content of a cell graph can be quantified using the *TLS embedding*, $\kappa = [\kappa_0, \ldots, \kappa_5] \in \mathbb{R}^6$ [69, 52]. This TLS-like organization metric considers only edges between B and T-cells and classifies them into several categories: $\alpha$ edges link two cells of the same type, while $\gamma_j$ edges connect a B to a T-cell, where $j$ is the number of B-cell neighboring the B-cell vertex (see Figure 4). Therefore, the entry $i$ of $\kappa$ is defined as the proportion of its $\gamma$ edges whose index is larger than $i$:

$$\kappa_i(G) = \frac{|E_{\text{BT}}| - |E_\alpha| - \sum_{j=0}^i |E_{\gamma_j}|}{|E_{\text{BT}}| - |E_\alpha|}, \tag{20}$$

where $|E_{\text{BT}}|$, $|E_\alpha|$, and $|E_{\gamma_j}|$ correspond to the number of edges whose both vertices are B or T-cells, of $\alpha$ edges and of $\gamma_j$ edges in a given graph, $G$. Note that, by definition, the entries of $\kappa$ take values between 0 and 1 and are monotonically non-increasing with $i$.

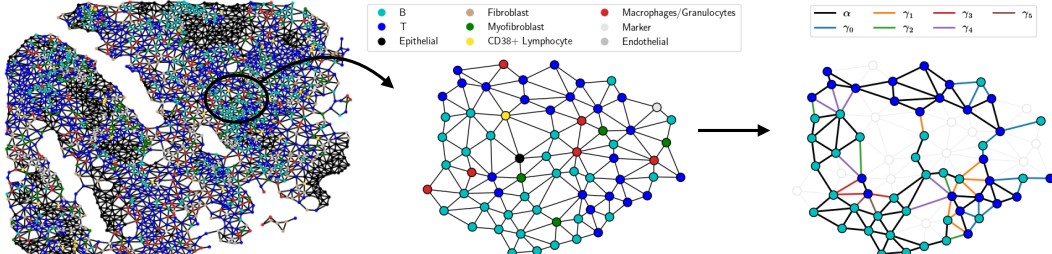

Figure 4: Extraction of a cell subgraph (center) from a WSI graph (left). From this cell subgraph, we can then compute the TLS embedding based on the classification of the edges into different categories, shown on the right. We can observe a cluster of B-cells surrounded by some support T-cells, characteristic of a high TLS content.

## F.4  Statistics of digital pathology datasets

In this section we provide the statistics for the low and high TLS content datasets. In Table 7, we provide their structural statistics and, in Table 8, the prevalence for each of the nine phenotypes (after mapping) across all the nodes in the datasets. In Figure 5, we provide their entry-wise distributions for the TLS embedding, $\kappa$.

Table 7: Digital pathology datasets statistics. Here, we report the same stats as in Table 5. #Train, #Val and #Test denote the number of graphs considered in the train, validation, and test splits, respectively.

| Dataset | Min. nodes | Max. nodes | Avg. nodes | Min. edges | Max. edges | Avg. edges | #Train | #Val | #Test |
|---|---|---|---|---|---|---|---|---|---|
| High TLS | 20 | 81 | 57.9 | 39 | 203 | 143.8 | 128 | 32 | 40 |
| Low TLS | 20 | 81 | 51.7 | 37 | 204 | 123.7 | 128 | 32 | 40 |

Table 8: Prevalence (in %) of the different cell phenotypes for the digital pathology datasets.

| Dataset | B | CD38+ Lymphocyte | Endothelial | Epithelial | Fibroblast | Macrophages/Granulocytes | Marker | Myofibroblast | T |
|---|---|---|---|---|---|---|---|---|---|
| High TLS | 39.3 | 1.9 | 4.6 | 9.4 | 4.4 | 6.3 | 0.6 | 7.2 | 26.4 |
| Low TLS | 7.7 | 2.4 | 5.9 | 33.4 | 17.7 | 8.4 | 0.2 | 9.9 | 14.1 |

## F.5  Baseline Method for Digital Pathology

The non-deep learning method used as baseline for the digital pathology dataset follows Madeira et al. [52]. This model learns three distributions by counting the frequencies of given events in the train dataset. In particular:

- Categorical distribution for the number of nodes, where the probability of sampling a given number of nodes is the same as its proportion in the train dataset, $D_{\text{train}}$, i.e.:

$$P(|X| = k) = \frac{|\{G \in D_{\text{train}} : |X| = k\}|}{|D_{\text{train}}|},$$

- Categorical distribution for the cell phenotypes, where the probability for each cell phenotype corresponds to its marginal probability in the dataset:

$$P(\text{Ph}(\nu) = \text{ph}_i) = \frac{\sum_{G \in D_{\text{train}}} |\{\nu \in X : \text{Ph}(\nu) = \text{ph}_i\}|}{\sum_{G \in D_{\text{train}}} |X|},$$

  where $\text{Ph}(\nu)$ refers to the phenotype of node $\nu$, and $\text{ph}_i$ denotes the specific phenotype labeled as $i$. These consist of the phenotypes described in Appendix F.2 with a fixed (arbitrary) order.

- Bernoulli distribution for the edge type (no edge *vs* edge) conditioned on the phenotypes of its two vertices, again computed based on its marginal distribution in the train set.

$$P(\text{Edge} \mid \text{Ph}(\nu_1) = \text{ph}_i, \text{Ph}(\nu_2) = \text{ph}_j) = \frac{\sum_{G \in D_{\text{train}}} |\{(v_1, v_2) \in C(G) : (v_1, v_2) \in E\}|}{\sum_{G \in D_{\text{train}}} |C(G)|},$$

  where $C(G) = \{v_1 \in X : \text{Ph}(v_1) = \text{ph}_i\} \times \{v_2 \in X : \text{Ph}(v_2) = \text{ph}_j\}$ for $1 \leq \text{ph}_i < \text{ph}_j \leq 9$.

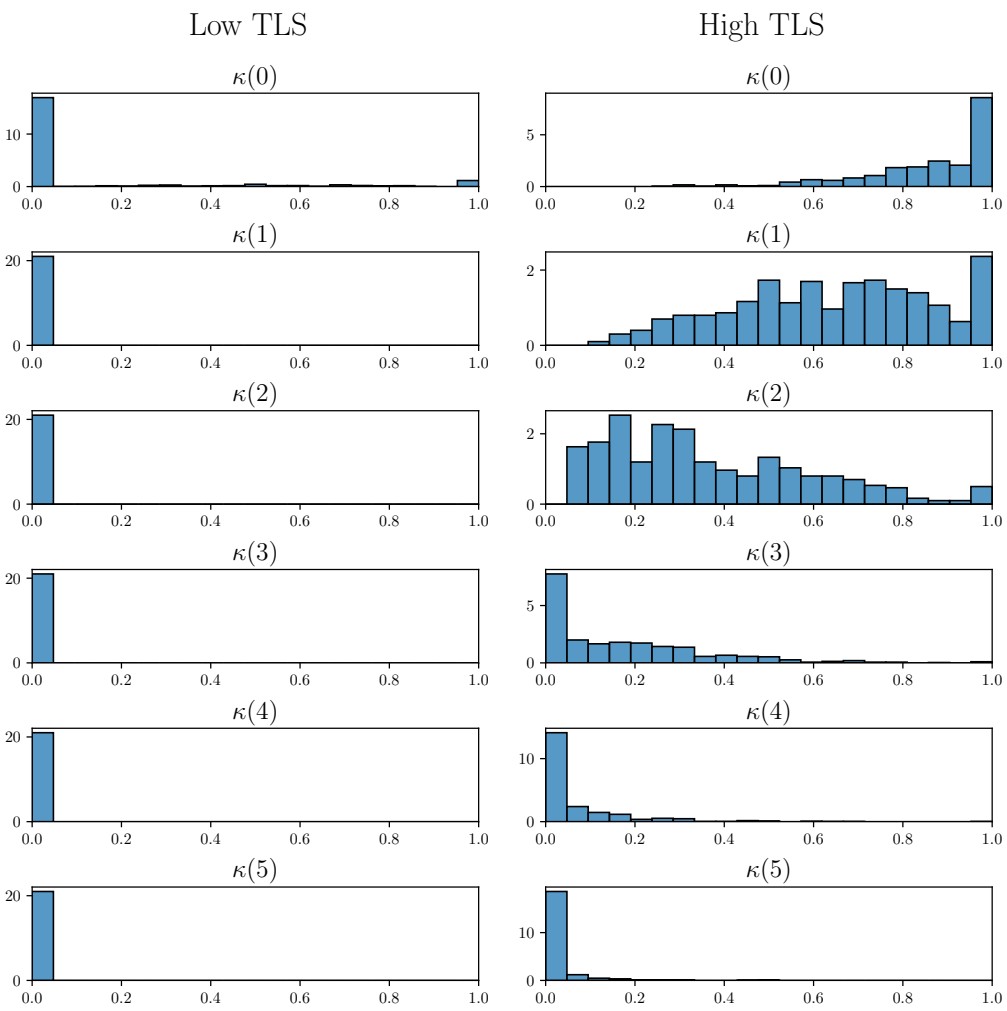

Figure 5: Distributions of the TLS embedding entries for the low TLS (left) and the high TLS (right) datasets.

To sample a new graph, we first sample a number of nodes for the graph from the first distribution. Then, for each of those nodes sample a cell phenotype from the second distribution. Finally, between every pair of cells, we sample an edge type given the two phenotypes previously sampled from the third distribution. This sampling algorithm is described in Algorithm 4.

---

**Algorithm 4:** Sampling Algorithm for the Digital Pathology Baseline

**Input:** Number of graphs to sample $N$

1  **for** $i = 1$ **to** $N$ **do**
2  $\quad$ Sample $|X| \sim P(|X|)$ ;$\qquad\qquad\qquad\qquad$ // Sample number of nodes
3  $\quad$ **for** $n = 1$ **to** $|X|$ **do**
4  $\quad\quad$ $X[n] \sim P(\mathrm{Ph}(\nu))$ ;$\qquad\qquad\qquad\qquad$ // Sample node phenotypes
5  $\quad$ **end**
6  $\quad$ **for** $1 \leq i < j \leq 9$ **do**
7  $\quad\quad$ $E[i, j] \sim P(\mathrm{Edge} \mid \mathrm{Ph}(X[i]), \mathrm{Ph}(X[j]))$ ;$\qquad\qquad$ // Sample edges
8  $\quad$ **end**
9  $\quad$ Store $G = (X, E)$;
10 **end**

---

# G   Molecular Datasets

## G.1   Exploring Planarity

In this section, we show the results for 3 molecular datasets: QM9 [81], MOSES [62] and GuacaMol [9]. Importantly, for QM9 and GuacaMol, we include formal charges as additional node labels, since this information has been shown beneficial for diffusion-based molecular generation [80]. For MOSES, such information is not available.

The metrics used to evaluate generation were:

- FCD - Fréchet ChemNet Distance [63], similar to Fréchet Inception Distance (FID) but for molecules, represented as SMILES. This metric evaluates the similarity between the generated and test molecule sets, providing an indicator for the sample quality in unconstrained settings;

- Uniqueness - proportion of not repeated molecules across all the generated molecules;

- Novelty - proportion of generated molecules that are not in the train set;

- Valid - proportion of generated molecules that are valid. This metric evaluates sample validity.

In particular, we explore planarity as target structural property in the molecular setting, as previously suggested by recent works in discriminative tasks [19]. Therefore, we also include the proportion of planar molecules from the generated set as an evaluation metric. As a remark, QM9 and MOSES are exclusively composed of planar molecules. GuacaMol contains 3 non-planar molecules out of 1273104 molecules in the train set and 3 non-planar molecules out of 238706 molecules in the test set. We considered these non-planar examples negligible in model training/evaluation, thus fully preserving the original dataset. The validation set has 79568 planar molecules. The results are shown in Table 9.

Table 9: Graph diffusion performance on molecular generation. The constraining property used for ConStruct is planarity. We performed five sampling runs for each method and present their results in the format mean ± standard error of the mean. For each run, we generated 10000, 25000, and 18000 generated molecules for QM9, MOSES, and GuacaMol, respectively, following the protocol from Vignac et al. [79]. Note that for MOSES and GuacaMol, we do not report their benchmarking metrics, as we focus on an overview comparative analysis of the two methods, DiGress+ and ConStruct. The FCD is computed using the official implementation from Preuer et al. [63].

| QM9 Dataset | | | | | |
|---|---|---|---|---|---|
| Model | FCD ↓ | Unique ↑ | Novel ↑ | Valid ↑ | Planarity ↑ |
| DiGress+ | $0.2090_{\pm0.0068}$ | $96.0_{\pm0.1}$ | $36.6_{\pm0.1}$ | $99.0_{\pm0.0}$ | $99.7_{\pm0.1}$ |
| ConStruct | $0.3443_{\pm0.0061}$ | $96.1_{\pm0.1}$ | $40.1_{\pm0.2}$ | $98.5_{\pm0.0}$ | $100.0_{\pm0.0}$ |

| MOSES Dataset | | | | | |
|---|---|---|---|---|---|
| Model | FCD ↓ | Unique ↑ | Novel ↑ | Valid ↑ | Planarity ↑ |
| DiGress+ | $0.5447_{\pm0.0080}$ | $100.0_{\pm0.0}$ | $93.5_{\pm0.1}$ | $87.5_{\pm0.1}$ | $100.0_{\pm0.0}$ |
| ConStruct | $0.6068_{\pm0.0045}$ | $100.0_{\pm0.0}$ | $93.7_{\pm0.1}$ | $84.1_{\pm0.1}$ | $100.0_{\pm0.0}$ |

| GuacaMol Dataset | | | | | |
|---|---|---|---|---|---|
| Model | FCD ↓ | Unique ↑ | Novel ↑ | Valid ↑ | Planarity ↑ |
| DiGress+ | $0.9663_{\pm0.0063}$ | $100.0_{\pm0.0}$ | $100.0_{\pm0.0}$ | $84.7_{\pm0.2}$ | $99.9_{\pm0.0}$ |
| ConStruct | $1.0538_{\pm0.0045}$ | $100.0_{\pm0.0}$ | $100.0_{\pm0.0}$ | $81.9_{\pm0.1}$ | $100.0_{\pm0.0}$ |

We only observe an incremental improvement in the planarity satisfaction of the output graphs of ConStruct, since DiGress+ learns to almost always generate planar graphs. As a consequence, the constrained model ends up being marginally less expressive than the unconstrained one, as extensively discussed in Appendix H.1. This impacts both sample quality and sample validity. In fact, while for the discriminative setting, the main aim is to design fully expressive architectures for classes of

graphs as broad as possible, e.g., PlanE [19] for planar graphs, in constrained generation this is not the case. As exemplified, planarity is too loose of a constraint, since the atoms composing molecules typically have low degrees and it becomes highly unlikely for the unconstrained diffusion model to violate planarity. This ends up slightly harming the performance of the constrained generative model without bringing the benefits of increased validity, as observed in Section 4.

As a side note for the interested reader, the low values of novelty for QM9 are a result of the nature of this dataset, which consists of an exhaustive enumeration of small molecules satisfying a given set of properties [78, 79]. Therefore, there is small room for the generation of new molecules within such space.

## G.2 Controlled Molecular Generation

In many real world scenarios, we want to generate molecules to target different goals, ranging from specific drug interactions to particular material properties. In such cases, we are not interested in generating any realistic molecules, but in obtaining molecules that are endowed with given properties matching our specific objectives. Constrained graph generation appears as a promising research direction to accomplish such tasks, as the generated molecules will necessarily verify the enforced properties by design. In this section, we explore how to use ConStruct to successfully address such challenge.

One relevant property of molecules is acyclicity. In molecules, this structural property dictates distinct chemical characteristics compared to their cyclic counterparts. In fact, acyclic molecules are frequently encountered in natural products and pharmaceuticals, where their linear structures contribute to enhanced solubility, bioavailability, and metabolic stability. Additionally, acyclic molecules offer simplified synthetic routes and reduced computational complexity in modeling studies.

We explore the generation of acyclic molecules by picking the absence of cycles as constraining property for ConStruct. Importantly, in contrast to all the experiments in the paper, we do *not* train with only graphs that verify the property. Instead, we use the two models trained in the unconstrained setting (from previous section): one with edge-absorbing transitions (ConStruct) and another with marginal edge transitions (DiGress+). We then sample from these models using the absorbing noise model and the projector for acyclicity. The results are presented in Table 10.

Table 10: Controlled graph diffusion for acyclic molecules. The constrained property used is acyclicity. Constrained DiGress+ denotes a model that was trained with a marginal noise model, but where the sampling is performed using the edge-absorbing noise model and projector. Both models were trained in the full QM9 dataset. We performed five sampling runs for each method and present their results in the format mean ± standard error of the mean. For each run, we generated 10000 molecules, following the protocol from Vignac et al. [79]. The FCD is computed using the official implementation from Preuer et al. [63].

| Model | QM9 Dataset | | | |
| | Unique ↑ | Novel ↑ | Valid ↑ | Acyclicity ↑ |
| --- | --- | --- | --- | --- |
| Constrained DiGress+ | 80.7 ±0.1 | 64.7 ±0.2 | 81.3 ±0.3 | 100.0 ±0.0 |
| ConStruct | 79.2 ±0.2 | 68.8 ±0.1 | 99.8 ±0.0 | 100.0 ±0.0 |

We observe that both methods output only acyclic molecules, which is a necessary consequence of the utilization of the projector. As the set of acyclic molecules is a subset of the set of unconstrained molecules, we verify some repetition among the generated samples. This leads to a decrease in the values of uniqueness for both models when compared to the unconstrained setting. Most remarkably, while the validity of the molecules generated by the model trained with the marginal noise model is significantly lower than the observed one for the unconstrained sampling setting, ConStruct preserves its high validity values (even higher than in the unconstrained setting). This result validates the foundation upon which ConStruct is laid: with the marginal noise model, the forward process distribution does not match the reverse one when employing the projector, harming the molecular validity of the generated instances. In contrast, the edge-absorbing noise model of ConStruct allows the forward and reverse processes to match, staying in distribution.

# H    Variants of ConStruct - Performance Analysis and Extensions

In this section, we perform some ablations to ConStruct/DiGress+ to further analyse its performance and explore methodological extensions to the proposed method.

## H.1    Performance Analysis

From Table 1, we observed that, for the specific case of the tree dataset, DiGress+ outperforms ConStruct. To explain why, we start by noting that approximately 97% of the graphs generated by DiGress+ already comply with the target structural property. This value indicates that DiGress+ had access to sufficient data and it is sufficiently expressive to learn the dependencies of tree graphs almost perfectly, leaving little room for improvement with ConStruct. In contrast, for the lobster and planar datasets, the corresponding values of DiGress+ are significantly lower, hinting the pertinence of ConStruct in such scenarios.

However, this observation alone does not explain the slight performance gap. To investigate further, we ran DiGress+ with an edge-absorbing noise model but without projector, designating it as DiGress+ [absorbing] in Table 11. We observe a significant decrease in performance with this modification. This suggests that it is the choice of the noise model that is hindering ConStruct's performance for this dataset. In fact, ConStruct outperforms DiGress+ [absorbing], emphasizing the relevance of the projector step. In any case, we remark that we did not adjust the variance schedule beyond the one proposed by Austin et al. [4], leaving room for potential improvement in this aspect. Importantly, we observe a dataset dependency of performance by applying the same modified model to the planar dataset (again, see Table 11), where it exhibits a significantly better average ratio compared to DiGress+. This observation aligns with recent research indicating that there is no clear evidence that an optimal noise model can be deduced *a priori* from dataset statistics [77].

## H.2    *A Posteriori* Modifications

An advantage of our setting is that the projector merely interferes with the sampling algorithm, avoiding to affect the efficiency of the diffusion model training. Otherwise, if we were to directly block the model's predictions, it would require a constraint satisfaction check for each potentially added edge at every forward pass, resulting in a prohibitive computational overhead.

In contrast, we could also consider the opposite setting: only applying *a posteriori* modifications to graphs generated by the unconstrained model. Two possible alternatives emerge:

- **DiGress+ [rejection]** - we reject the final samples generated by the unsconstrained model that do not satisfy the provided constraint. While we should expect a good performance from this approach, it wastes computational resources as it requires discarding the rejected graphs and restarting the whole sampling process until we get the desired amount of generated graphs.

- **DiGress+ [projection]** - we only apply the projector to the final samples generated by the unconstrained method. For example, in the case of planarity as target property, we could find the maximal planar subgraphs of the generated samples. This method would necessarily provide a more efficient sampling procedure (as we only execute the projector step once).

We provide the results in synthetic graphs for both methods in Table 11. We observe that DiGress+ [rejection] attains great V.U.N. values, as expected. Nevertheless, we analyse its alarming computational inefficiencies in Appendix D.3. Additionally, we see that DiGress+ (projection) achieves worse performance than ConStruct. We attribute this result to the fact that such a scheme fails to inform the generative model about the constraining condition throughout the reverse process, thus not harnessing the full expressivity of the diffusion model. We also attribute the anomalously good performance of this method for the tree dataset to the optimal properties of the projector in such case (see Theorem 2).

Finally, considering the two extreme cases described above (blocking edges at every forward pass *vs a posteriori* modifications), we conclude that ConStruct finds itself in a sweet spot in the trade-off between additional computational burden and constraint integration into the generative process.

## H.3 Likelihood-based constrained generation

We also consider two variants of ConStruct where the projector is no longer independent from the diffusion model. Instead, we use the associated likelihoods to each of the sampled candidate edges at a given time step $t$, $p_\theta(e_{ij}^{t-1}|G^t)$, to define an order by which we add the edges. Therefore, instead of uniformly sampling them at random, we propose two methods of integrating such information:

- Deterministic: we add the edges with higher $p_\theta(e_{ij}^{t-1}|G^t)$ first.
- Stochastic: we sample without replacement from the set of candidates edges, where the probability of sampling each of them is proportional to the respective $p_\theta(e_{ij}^{t-1}|G^t)$.

We present the results for these likelihood-based variants of ConStruct in Table 11. As we can observe, for all the analysed datasets, the three ConStruct variants are statistically equivalent in terms of performance. In terms of efficiency, there is no meaningful difference among the three methods: even though the uniformly random sampling does not have to access $p(e_{ij}^{t-1}|G^t)$, nor sort the order of the corresponding edges, the computational overburden of these operations is negligible. We remark that the theoretical analysis performed in Appendix B also holds for the stochastic likelihood-based variant, but not for the deterministic one, as the latter is not able to select any permutation from the candidate edges with non-zero probability: up to the degenerate cases where different candidate edges have the same $p(e_{ij}^{t-1} \mid G^t)$, it always selects candidate edges by the same order.

Table 11: Graph generation performance on synthetic graphs. DiGress+, ConStruct are retrieved from Table 1, from which we follow the same experimental protocol. DiGress+ [absorbing] denotes DiGress+ with an edge-absorbing noise model. DiGress+ [rejection] refers to the baseline that rejects the unconstrainedly generated graphs that do not satisfy the constraint and resamples until the intended number of valid graphs is reached, while DiGress+ [projection] directly applies the projector on unconstrainedly generated graphs. ConStruct [model - det] and ConStruct [model - stoch] denote the deterministic and stochastic likelihood-based variants of ConStruct.

| Model | Deg. ↓ | Clus. ↓ | Orbit ↓ | Spec. ↓ | Wavelet ↓ | Ratio ↓ | Valid ↑ | Unique ↑ | Novel ↑ | V.U.N. ↑ | Property ↑ |
|---|---|---|---|---|---|---|---|---|---|---|---|
| | | | | | Planar Dataset | | | | | | |
| Train set | 0.0002 | 0.0310 | 0.0005 | 0.0038 | 0.0012 | 1.0 | 100 | 100 | 0.0 | 0.0 | 100 |
| DiGress+ | 0.0008 ±0.0001 | 0.0410 ±0.0033 | 0.0048 ±0.0004 | 0.0056 ±0.0004 | 0.0020 ±0.0002 | 3.6 ±0.2 | 76.4 ±1.3 | 100.0 ±0.0 | 100.0 ±0.0 | 76.4 ±1.3 | 76.4 ±1.3 |
| ConStruct | 0.0003 ±0.0001 | 0.0403 ±0.0047 | 0.0004 ±0.0001 | 0.0053 ±0.0004 | 0.0009 ±0.0001 | 1.1 ±0.1 | 100.0 ±0.0 | 100.0 ±0.0 | 100.0 ±0.0 | 100.0 ±0.0 | 100.0 ±0.0 |
| DiGress+ [absorbing] | 0.0006 ±0.0002 | 0.0383 ±0.0041 | 0.0028 ±0.0005 | 0.0050 ±0.0002 | 0.0010 ±0.0001 | 2.4 ±0.2 | 42.4 ±1.0 | 100.0 ±0.0 | 100.0 ±0.0 | 42.4 ±1.0 | 42.4 ±1.0 |
| DiGress+ [rejection] | 0.0008 ±0.0001 | 0.0418 ±0.0035 | 0.0022 ±0.0001 | 0.0054 ±0.0004 | 0.0019 ±0.0001 | 2.6 ±0.2 | 100.0 ±0.0 | 100.0 ±0.0 | 100.0 ±0.0 | 100.0 ±0.0 | 100.0 ±0.0 |
| DiGress+ [projection] | 0.0003 ±0.0001 | 0.0347 ±0.0030 | 0.0013 ±0.0002 | 0.0056 ±0.0003 | 0.0015 ±0.0001 | 1.6 ±0.1 | 100.0 ±0.0 | 100.0 ±0.0 | 100.0 ±0.0 | 100.0 ±0.0 | 100.0 ±0.0 |
| ConStruct [model - det] | 0.0004 ±0.0001 | 0.0416 ±0.0040 | 0.0005 ±0.0002 | 0.0050 ±0.0003 | 0.0009 ±0.0001 | 1.3 ±0.2 | 100.0 ±0.0 | 100.0 ±0.0 | 100.0 ±0.0 | 100.0 ±0.0 | 100.0 ±0.0 |
| ConStruct [model - stoch] | 0.0004 ±0.0001 | 0.0404 ±0.0038 | 0.0005 ±0.0002 | 0.0050 ±0.0003 | 0.0009 ±0.0001 | 1.2 ±0.1 | 100.0 ±0.0 | 100.0 ±0.0 | 100.0 ±0.0 | 100.0 ±0.0 | 100.0 ±0.0 |
| | | | | | Tree Dataset | | | | | | |
| Train set | 0.0001 | 0.0000 | 0.0000 | 0.0075 | 0.0030 | 1.0 | 100 | 100 | 0.0 | 0.0 | 100 |
| DiGress+ | 0.0002 ±0.0001 | 0.0000 ±0.0000 | 0.0000 ±0.0000 | 0.0092 ±0.0005 | 0.0032 ±0.0001 | 1.3 ±0.2 | 91.6 ±0.7 | 100.0 ±0.0 | 100.0 ±0.0 | 91.6 ±0.7 | 97.0 ±0.8 |
| ConStruct | 0.0003 ±0.0001 | 0.0000 ±0.0000 | 0.0000 ±0.0000 | 0.0073 ±0.0008 | 0.0034 ±0.0002 | 1.9 ±0.3 | 83.0 ±1.8 | 100.0 ±0.0 | 100.0 ±0.0 | 83.0 ±1.8 | 100.0 ±0.0 |
| DiGress+ [absorbing] | 0.0004 ±0.0002 | 0.0000 ±0.0000 | 0.0000 ±0.0000 | 0.0079 ±0.0006 | 0.0034 ±0.0002 | 2.3 ±0.5 | 72.8 ±0.6 | 100.0 ±0.0 | 100.0 ±0.0 | 72.8 ±0.6 | 85.6 ±1.4 |
| DiGress+ [rejection] | 0.0002 ±0.0001 | 0.0000 ±0.0000 | 0.0000 ±0.0000 | 0.0093 ±0.0004 | 0.0032 ±0.0000 | 1.4 ±0.3 | 100.0 ±0.0 | 100.0 ±0.0 | 100.0 ±0.0 | 100.0 ±0.0 | 100.0 ±0.0 |
| DiGress+ [projection] | 0.0002 ±0.0001 | 0.0000 ±0.0000 | 0.0000 ±0.0000 | 0.0092 ±0.0004 | 0.0031 ±0.0001 | 1.3 ±0.2 | 94.0 ±0.3 | 100.0 ±0.0 | 100.0 ±0.0 | 94.0 ±0.3 | 100.0 ±0.0 |
| ConStruct [model - det] | 0.0003 ±0.0001 | 0.0000 ±0.0000 | 0.0000 ±0.0000 | 0.0076 ±0.0008 | 0.0034 ±0.0001 | 1.9 ±0.3 | 83.2 ±1.7 | 100.0 ±0.0 | 100.0 ±0.0 | 83.2 ±1.7 | 100.0 ±0.0 |
| ConStruct [model - stoch] | 0.0004 ±0.0001 | 0.0000 ±0.0000 | 0.0000 ±0.0000 | 0.0072 ±0.0008 | 0.0034 ±0.0001 | 1.9 ±0.3 | 83.2 ±1.7 | 100.0 ±0.0 | 100.0 ±0.0 | 83.2 ±1.7 | 100.0 ±0.0 |
| | | | | | Lobster Dataset | | | | | | |
| Train set | 0.0002 | 0.0000 | 0.0000 | 0.0070 | 0.0070 | 1.0 | 100 | 100 | 0.0 | 0.0 | 100 |
| DiGress+ | 0.0005 ±0.0001 | 0.0000 ±0.0000 | 0.0000 ±0.0000 | 0.0114 ±0.0006 | 0.0093 ±0.0005 | 1.8 ±0.1 | 79.0 ±1.1 | 98.0 ±0.7 | 96.6 ±0.6 | 69.4 ±1.2 | 76.8 ±1.7 |
| ConStruct | 0.0003 ±0.0001 | 0.0000 ±0.0000 | 0.0000 ±0.0000 | 0.0092 ±0.0009 | 0.0074 ±0.0004 | 1.3 ±0.2 | 86.8 ±2.4 | 98.8 ±0.6 | 97.0 ±0.9 | 83.2 ±2.3 | 100.0 ±0.0 |
| DiGress+ [rejection] | 0.0006 ±0.0001 | 0.0000 ±0.0000 | 0.0000 ±0.0000 | 0.0130 ±0.0010 | 0.0106 ±0.0006 | 2.1 ±0.2 | 100.0 ±0.0 | 96.4 ±0.2 | 95.6 ±0.8 | 93.6 ±0.8 | 100.0 ±0.0 |
| DiGress+ [projection] | 0.0006 ±0.0001 | 0.0000 ±0.0000 | 0.0000 ±0.0000 | 0.0109 ±0.0006 | 0.0098 ±0.0005 | 2.0 ±0.1 | 77.8 ±1.2 | 98.2 ±0.6 | 96.6 ±0.6 | 73.6 ±1.0 | 100.0 ±0.0 |
| ConStruct [model - det] | 0.0003 ±0.0001 | 0.0000 ±0.0000 | 0.0000 ±0.0000 | 0.0093 ±0.0008 | 0.0075 ±0.0003 | 1.2 ±0.1 | 87.0 ±2.3 | 98.8 ±0.6 | 97.0 ±0.9 | 83.4 ±2.2 | 100.0 ±0.0 |
| ConStruct [model - stoch] | 0.0003 ±0.0001 | 0.0000 ±0.0000 | 0.0000 ±0.0000 | 0.0093 ±0.0008 | 0.0075 ±0.0003 | 1.2 ±0.1 | 87.0 ±2.3 | 98.8 ±0.6 | 97.0 ±0.9 | 83.4 ±2.2 | 100.0 ±0.0 |

# I  Visualizations

In this section, we provide several visualizations of the final generated graphs, comparing them to the ones observed in the different datasets. We also visually expose the effect of the projector in different timesteps.

## I.1  Graphs generated by ConStruct

Here we visually compare the graphs from the different datasets to the ones generated by ConStruct.

### I.1.1  Synthetic Datasets

We provide plots of the sampled graphs from ConStruct for the different datasets: planar in Figure 6, tree in Figure 7, and lobster in Figure 8.

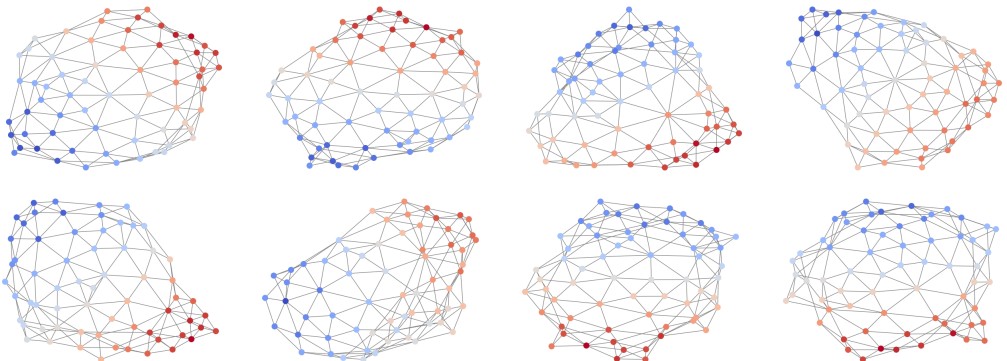

Figure 6: Uncurated set of dataset graphs (top) and generated graphs by ConStruct (bottom) for the planar dataset.

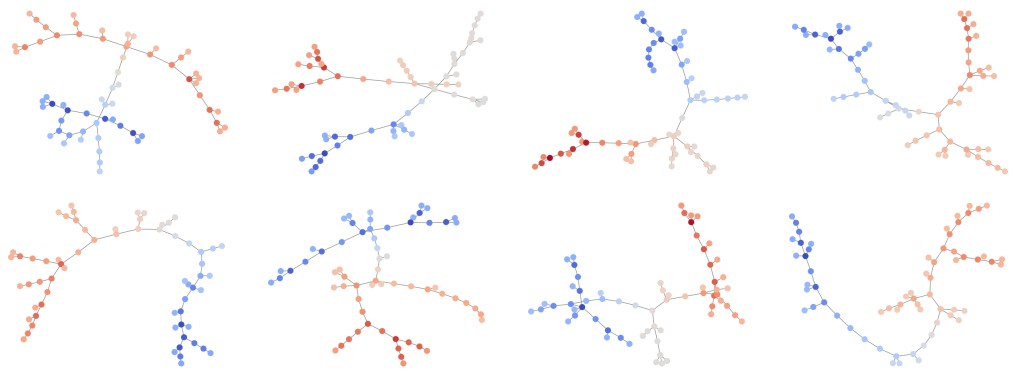

Figure 7: Uncurated set of dataset graphs (top) and generated graphs by ConStruct (bottom) for the tree dataset.

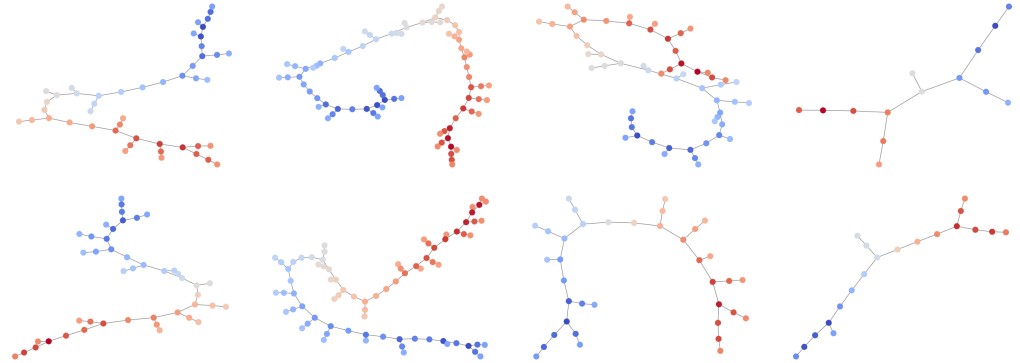

Figure 8: Uncurated set of dataset graphs (top) and generated graphs by ConStruct (bottom) for the lobster dataset.

### I.1.2 Digital Pathology

We provide plots of the sampled graphs from ConStruct for the low TLS content dataset in Figure 9 and for the high TLS content dataset in Figure 10. We also provide several snapshots throughout the reverse process of ConStruct in Figure 11 to illustrate it as an edge insertion procedure.

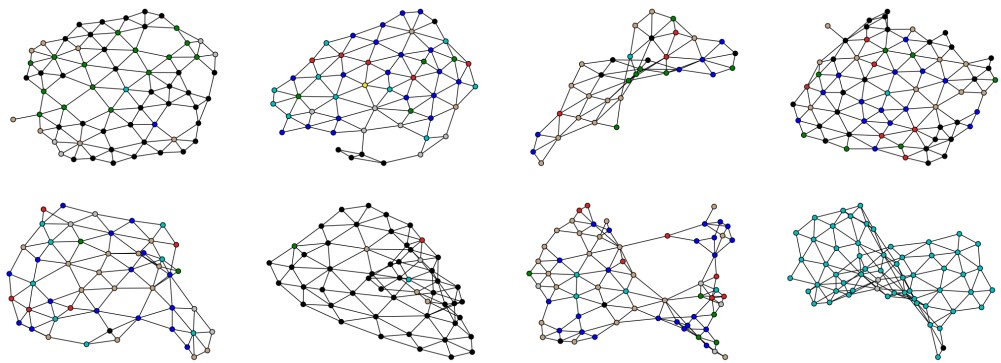

Figure 9: Uncurated set of dataset graphs (top) and generated graphs by ConStruct (bottom) for the low TLS dataset. The phenotype color key is presented in Figure 4.

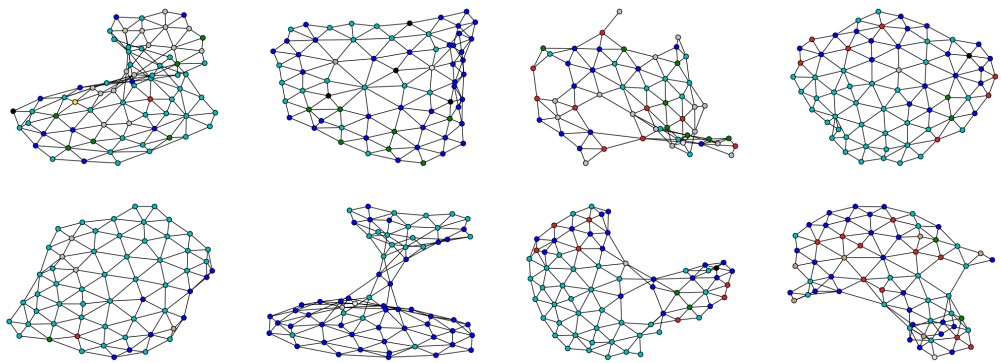

Figure 10: Uncurated set of dataset graphs (top) and generated graphs by ConStruct (bottom) for the high TLS dataset. The phenotype color key is presented in Figure 4.

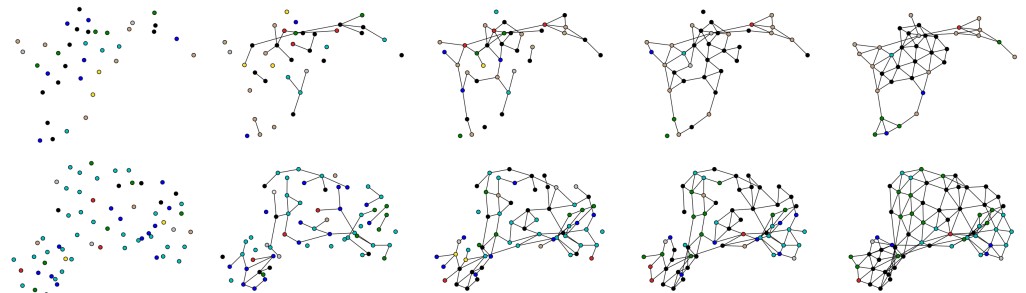

Figure 11: Reverse processes for generation of low (top) and high (bottom) TLS content graphs using ConStruct. We start from a graph without any edge on the left ($t = T$) and progressively build the graph, as a consequence of the absorbing noise model. The node types switch along the trajectory due to the marginal noise model. On the right, we have a fresh new sample ($t = 0$). The phenotypes color key is presented in Figure 4.

### I.2 Visualizing Intermediate Graphs (Before and After Projector)

In this section, we provide some visualizations of intermediate graphs obtained throughout the reverse process for three different datasets: planar, tree, and lobster. In Figure 12, we highlight the effect of the projector in rejecting the candidate edges that lead to property violation.

## J  Impact Statement

The primary objective of this paper is to enhance graph generation methodologies by enabling the integration of hard constraints into graph diffusion models. Although this problem holds significance for several real-world applications, including digital pathology and molecular generation, as exemplified in the paper, as well as protein design, the potential implications extend to advances in biomedical and chemical research. This development has the capacity to yield both positive and negative societal outcomes. Nonetheless, despite the potential for real-world impact, we currently do not identify any immediate societal concerns associated with the proposed methodology.

For the particular case of the digital pathology setting, while the generated graphs are able to mimic clinically relevant structures, they remain too small to have any direct clinical impact. Pathologists use whole-slide images for informed decisions, whose corresponding cell graphs typically comprise a total number of nodes 3 to 4 orders of magnitude above the graphs generated at this stage.

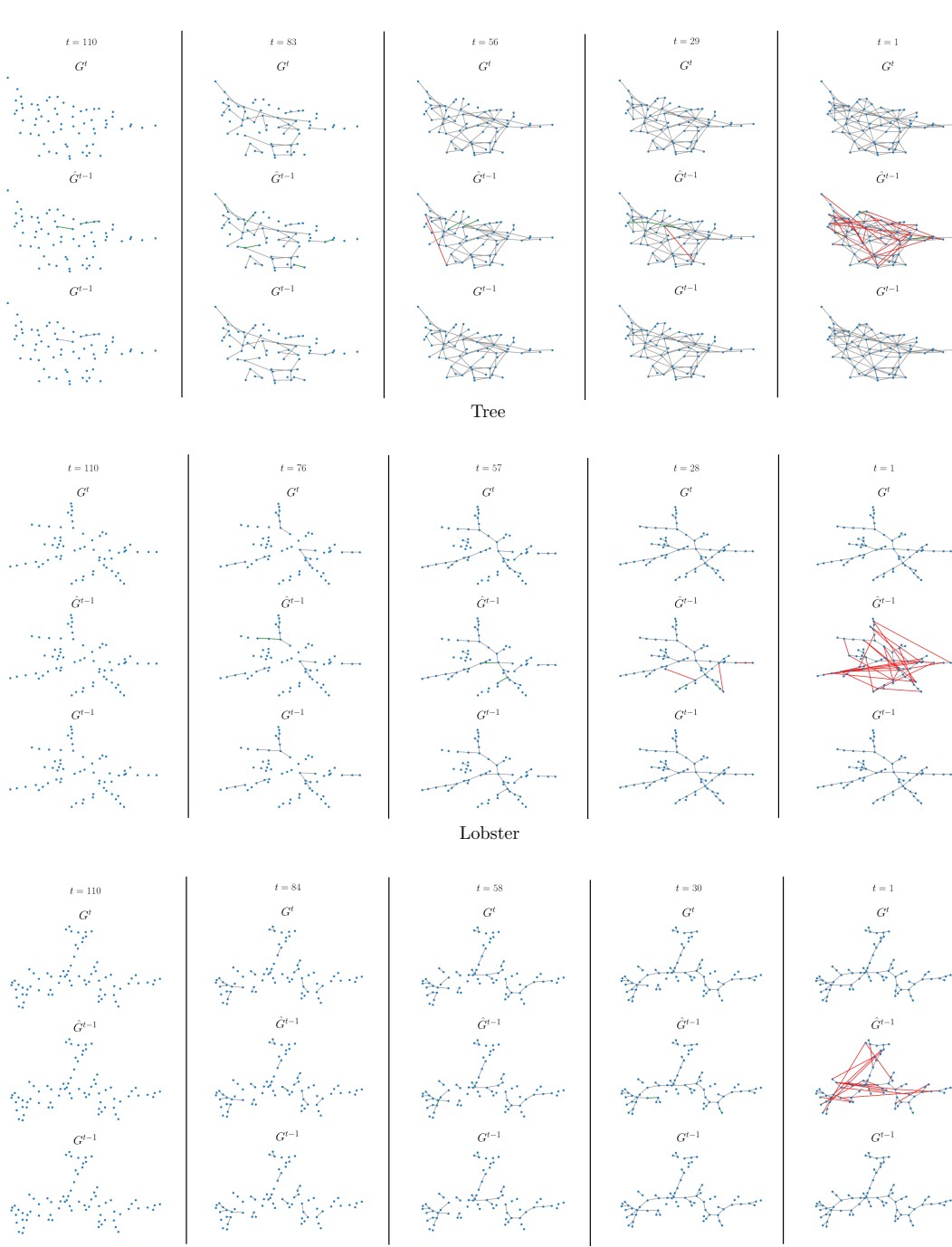

Figure 12: Visualizations of intermediate graphs throughout the reverse process. The notation follows the one of the rest of the paper: we obtain $G^{t-1}$ after applying the projector on $\hat{G}^{t-1}$, which in turn is obtained from $G^t$ through the diffusion model. From the new edges obtained in $\hat{G}^{t-1}$, we color them in green when they do not break the constraining property and in red otherwise. We can observe that the red edges are rejected. To better emphasize the edge rejection by the projector, we do not use a fully trained model and use a trajectory length, $T$, smaller than usual, resulting in less accurate edge predictions.

