# OpenReview forum: "Generative Modelling of Structurally Constrained Graphs"
_NeurIPS.cc/2024/Conference — NeurIPS 2024 poster_

### Official Review · Reviewer_FDH7 · 2024-06-13

**Soundness:** 3
**Presentation:** 4
**Contribution:** 4
**Rating:** 7
**Confidence:** 4

**Summary:**

The authors proposed ConStruct, a graph generative framework that enables hard-constraining graph topological properties that hold upon edge deletion throughout the entire sampling trajectory. Specifically, the authors model the forward (data-to-prior) process using an edge absorbing noise model, and they predict the reversed edge insertion process using a GNN projector.

**Strengths:**

**S1.** The proposed method is the first graph constrained discrete generative framework. The hard-constraining graph generation task is challenging and meaningful.

**S2.** The authors provided theoretical guarantee for the generation quality of ConStruct.

**S3.** The authors improve the sampling efficiency using the incremental constraint satisfaction algorithm in the spirit of curriculum learning.

**Weaknesses:**

**W1.** The ConStruct is only applicable to edge-deletion invariant properties (Definition 3.1). Thus, it cannot be applied to hard-constrain more complicated and general graph properties, e.g. chemical properties for molecules.

However, this limitation does NOT weaken the contribution of this paper and should be considered a future research direction of hard-conditioned graph generation.

**W2.** The projector module (Figure 2 and Algorithm 3) introduces intractability in likelihood estimation of the graph, making it hard to estimate the likelihood of the graph instances generated by ConStruct.

**W3.** ConStruct seems to be a autoregressive method, which means that the sampling complexity is in proportional to the graph size. In contrast, the diffusion-based method generates the whole graph and keeps refining it along the generative trajectory. Thus, it seems that ConStruct takes more generative steps for large graphs (w.r.t the amount of nodes and edges) when compared to diffusion-based models. I recommend the authors to assess the time complexity and efficiency of this model compared to diffusion-based methods.

**W4.** It seems that the authors did not compare their method against the two mentioned (Page 3, line 88-89) discrete-diffusion graph generative models EDGE [1] and Graph-ARM [2]. By the way, I recommend the authors to cite the latest or published version rather than the ArXiv versions.

[1]. Xiaohui Chen, Jiaxing He, Xu Han, and Li-Ping Liu. 2023. Efficient and degree-guided graph generation via discrete diffusion modeling. In Proceedings of the 40th International Conference on Machine Learning (ICML'23), Vol. 202. JMLR.org, Article 181, 4585–4610.

[2]. Kong, L., Cui, J., Sun, H., Zhuang, Y., Prakash, B.A., & Zhang, C. (2023). Autoregressive Diffusion Model for Graph Generation. *International Conference on Machine Learning*.

**Questions:**

**Q1.** Please clarify the definition of $\Delta^b$ in Page 5, line 171.

**Q2.** During the reverse sampling process, how to guarantee the existence of a feasible $G^{t-1}$ among the candidates induced by discarding some newly added edges from $G^t$ to $\widehat{G}^{t-1}$ (or equivalently, the intersection of $\mathcal{C}$ and $\mathcal{G}^{t-1}$ in Theorem 1 is not empty)? If there is no feasible candidate, will the generative procedure be prematurely existed?

**Q3.** Following **W2**, I notice that there is a seemingly discrepancy between the training and sampling process. The effect of the projector is absent during the training process. I hope the authors can add some analysis and clarifications on why the edge predictor trained without projector can still provide satisfactory samples.

**Limitations:**

The authors adequately discussed the limitations of the proposed method.

They discuss in detail the performance limitations of their approach in Appendix H.1. They also point out potential further improvements and extensions to their approach in Section 5. Finally, they discuss computational efficiency and scalability of the proposed method in Section 3.5 and Appendices D.2 and D.3.

---

> ### Author Rebuttal · Authors · 2024-08-06
>
> We thank the reviewer for acknowledging the importance of the topic under consideration and for the insightful comments. We address the reviewer's concerns in the following points:
>
> **W1**: We agree with the reviewer and do also envision the extension to joint node-edge constraints, e.g. valency constraints in molecular generation, as an exciting future direction. We extend on this topic on item 3 of the global rebuttal.
>
> **W2**: We remark that, despite the lack of a tractable evidence lower bound for the likelihood, the proposed projector pushes ConStruct to remarkable empirical performance, outperforming unconstrained models.
>
> **W3**: Even though ConStruct allows for constrained generation in a similar manner to autoregressive models, it is actually a diffusion-based framework.
> We thoroughly analyse its complexity and efficiency in section 3.5 and appendix C. Based on this, we provide a more explicit comparison in terms of complexity to the underlying discrete diffusion methods in item 1 of the global rebuttal. From such comparison, we are able to conclude that ConStruct is a scalable method.
>
> **W4**: For EDGE, we do compare for planar and tree datasets (Table 1). We did not test it in the digital pathology setting because we only picked "the methods that, besides DiGress, attain non-zero V.U.N. for the planar dataset in Table 1, which we consider as a proxy for performance in the digital pathology datasets due to the structural similarities between the datasets" (ll. 310-312, pg. 9). In fact, the digital pathology dataset is even more challenging since it contains attributed planar graphs.
>
> For GraphARM, unfortunately the authors do not report results on any of the used datasets, neither provide their code (to the best of our knowledge), which makes meaningful comparisons difficult.
>
> We appreciate the reviewer's suggestion to update citations and have addressed this.
>
> **Q1**: $\Delta^k$ denotes the $k$-simplex:
>
> $$
> \left\lbrace \left(\lambda_0, \lambda_1, \ldots, \lambda_k\right) \in \mathbb{R}^{k+1} \mid \lambda_i \geq 0 \text{ for all } i,  \text{ }  \sum_{i=0}^k \lambda_i = 1 \right\rbrace.
> $$
>
> We added this clarification to the notation paragraph and we corrected its usage in l.171 accordingly (to $\Delta^{b-1}$).
>
> **Q2**: If all the newly proposed edges (inserted from $G^t$ to $\hat{G}^{t-1}$) by the diffusion model lead to constraint violation, then the graph remains the same as in the previous step, $G^{t-1}=G^t$ (so the intersection of $\mathcal{C}$ and $\mathcal{G}^{t-1}$ is trivially non-empty). We do not early exit the reverse process in such case because the denoising network takes as input both $G^t$ and the timestep $t$. Therefore, even if for a given timestep $t$ all the proposed edges are rejected, the model predicted probabilities can change in the next reverse step, since the model has now a different input (still the same graph $G^{t-1}$($=G^t$) but the timestep is $t-1$). Note that the case where the diffusion model does not propose any new edge (as we dot not lower nor upper bound the number of edge insertions at each step) falls within that case as well.
>
> Additionally, we remark that the diffusion model outputs a probability distribution over graphs, from which we sample the proposed graph. As a result, the proposed edges at each reverse step are not deterministic, even with the same inputs (thus outputs) to the model.
>
> **Q3**: In training, as the edge-absorbing noise model only removes edges, the training noisy graphs necessarily remain within the domain of graphs that satisfy the desired edge-deletion invariant property. Therefore, the denoising network is only trained for graphs that satisfy the desired property.
>
> In sampling, even though the posterior term of the edge absorbing noise model ensures the reverse as an edge insertion process (this is a direct implication from using eq. 6 in eq. 5), it does not guarantee that the successive graphs remain within the domain of graphs which satisfy the desired property.
> In fact, since the denoising neural network is not a perfect edge predictor and the edges are sampled independently, sometimes we obtain intermediate graphs that are outside of such domain, inducing out-of-distribution prediction to the denoising neural network for the next step. To counter this, the projector is used to push back the graph generative trajectory to the domain where the model was actually trained. Note that the projector is only used to correct the trajectory when it gets out of the desired domain and does not interfere whenever the trajectory is within domain. Therefore, the projector is actually reducing the discrepancy between training and sampling processes and promoting in-distribution prediction for the denoising neural network, resulting in improved performance both in terms of sample quality and validity.
>
> These results are aligned with the intuition provided in other constrained diffusion models for continuous state-spaces that also empirically validated the benefit of promoting the match of distributions between training and sampling [1,2,3].
>
> [1] - Lou, Aaron, and Stefano Ermon. "Reflected diffusion models." ICML, 2023.
>
> [2] - Fishman, Nic, et al. "Metropolis Sampling for Constrained Diffusion Models." NeurIPS, 2024.
>
> [3] - Liu, Guan-Horng, et al. "Mirror diffusion models for constrained and watermarked generation." NeurIPS, 2024.

---

> > ### Comment · Reviewer_FDH7 · 2024-08-09
> >
> > Thanks for the detailed responses. I hope the authors to thoroughly scrutinize the manuscript to ensure the notation system is correct and self-contain.
> >
> > I raise my score to 7 because my Q2 is fully addressed. The setting is new and important, the proposed method is sounded. The logic flow of the paper is smooth. The theoretical analysis is adequate.
> >
> > I cannot give a higher rate because this paper is restricted to 'edge-deletion-invariant constraints', which is not general enough.

---

> > > ### Author Response · Authors · 2024-08-09
> > >
> > > We thank the reviewer for their time and the score update. We remain open to any further clarifications.

---

### Official Review · Reviewer_ZwhZ · 2024-07-08

**Soundness:** 3
**Presentation:** 3
**Contribution:** 3
**Rating:** 6
**Confidence:** 4

**Summary:**

The paper presents a novel diffusion model, ConStruct, to generate graphs that follow certain pre-specified properties. ConStruct involves an edge-absorbing forward process and a projected edge-addition reverse process to sample graphs that satisfy pre-specified constraints. The novelty of their method comes from a simple random-sampling-based projection algorithm that samples constrained graphs at each diffusion step. Experimental results show that ConStruct can generate more realistic synthetic graphs that have a pre-defined constraint for the whole domain. They also show its applicability to real-world pathology graphs, by leveraging the fact that they tend to be planar.

**Strengths:**

- The paper is well-written and easy to read.
- Experimental results are comprehensive for the synthetic graph datasets and use all representative metrics.
- The paper also provides results of a real-world pathology dataset to highlight how structurally constraining the diffusion model is useful once we can establish some structure on the real-world graphs. In particular, they consider the planarity of the breast cancer pathology cell graphs.
- ConStruct provides efficient ways to solve an otherwise NP-hard problem of projecting to hard constraints of acyclicity and planarity. The sampling time of ConStruct is at par with its unconstrained counterparts, which is quite impressive.

**Weaknesses:**

- It is not clear how applicable such domain-level constraints will be for real-world graphs that do not have a well-defined constraint. While the motivating example of pathology cell graphs is appreciated, it is difficult to see how it generalizes.
- The idea is similar to PRODIGY which can be applied to any diffusion model and constraint including discrete models in the latest version [1] (which can be ignored but worthwhile to mention). Upon ignoring the minor requirement of an edge-absorbing forward process, ConStruct can be seen as an approximation of the projection operator where the distance is calculated as a GED instead of the Euclidean distance between the adjacency matrices. Thus, the novelty of the proposed method is limited.
- Furthermore, the authors claim that PRODIGY distorts the underlying diffusion process even though PRODIGY takes a fractional step to the closest noisy graph that satisfies the given constraint. On the other hand, ConStruct finds a random graph that satisfies the constraint and takes a full step in that direction. Thus, it seems intuitively that ConStruct distorts the process more than PRODIGY, as opposed to the authors' claims.
- The major novelty of the method comes from the proposed projection algorithm that tries to circumvent the NP-hardness by iteratively adding a random edge if it satisfies the constraint, which can be seen as a simple randomized greedy algorithm, which is quite well-studied in the discrete optimization literature. Theorem 1 in this vein is a bit trivial since it is expected that the optimal graph will belong to a randomly-edited set but it is not clear how easily it will be sampled for an arbitrary constraint. In the absence of this analysis, the proposed projection algorithm is not suitable for application.
- While the authors compare against SPECTRE, they do not provide an elaborate discussion with this important related work that proposed this problem of including structurally-constrained generative models.
- The proposed framework is limited to edge-deletion invariant (or more formally, downward-closed) constraints while the PRODIGY framework can theoretically handle a larger range of constraints including box constraints.
- ConStruct is limited to non-attributed structural constraints due to the edit distance formulation and thus, cannot be applied to important molecular constraints.

[1] Sharma, Kartik, Srijan Kumar, and Rakshit Trivedi. "Diffuse, Sample, Project: Plug-And-Play Controllable Graph Generation." Forty-first International Conference on Machine Learning. 2024

**Questions:**

See above weaknesses.

**Limitations:**

The authors do not adequately discuss the limitations of their work and the potential negative impacts of their work. It will be really useful to elaborate on the limitations of their work, e.g., limited to structural constraints of a certain kind. Furthermore, since they are using a pathology dataset of breast cancer studies, the authors must discuss the potential negative societal impacts of their analysis, particularly when the "planar" generated graphs are used as a data augmentation tool for the downstream application for cancer detection or such.

---

> ### Author Rebuttal · Authors · 2024-08-06
>
> We appreciate the detailed comments and pertinent questions. We address the raised concerns below:
>
> **W1**: We acknowledge that our model is tailored for the task of constrained generation, which requires the explicit and unambiguous definition of the constraint, as reflected in the paper's title. Despite not universal, we believe our framework is quite general. It proves effective for various properties in structurally constrained synthetic graphs and digital pathology, outperforming unconstrained models, and can also be used to guide unconstrained models (e.g., generating acyclic molecules, App. G.2). Additionally, we (non-exhaustively) enumerate several real-world applications beyond digital pathology where essential constraints are well-defined and edge-deletion invariant (l. 144-147, pg 4).
>
> **W2**: Both works indeed address similar tasks. However, they differ fundamentally: PRODIGY offers efficient guidance of pre-trained models by relaxing adjacency matrices to continuous spaces (implicitly imposing an order between states) and finding low overhead projections there. Yet, it does not guarantee constraint satisfaction and faces a trade-off between performance and constraint satisfaction due to mismatched train/sample distributions. In contrast, ConStruct, while also suitable for guided generation (App. G.2), is designed for constrained generation. It treats adjacency matrices as discrete, where there are no efficient solutions for projections into arbitrary subclasses of graphs (e.g. maximum planar subgraph is NP-hard). This results from the lack of inherent ordering between states and the combinatorial nature of the domains of valid graphs. Nonetheless, ConStruct ensures constraint satisfaction with matched distributions, thereby improving performance over unconstrained models, and maintains efficiency through incremental algorithms and blocking edge hash tables. Thus, ConStruct is not an approximation of continuous-space projections and we believe it to be a full contribution in itself.
>
> **W3**: Our mention to the potentially compromised smoothness of the diffusion process refers to the thresholding step required by PRODIGY to convert the continuous adjacency matrix back to a discrete one, either at the end of the diffusion process (continuous diff.) or at each reverse step (discrete diff., a variant not available at submission). This step can disrupt smoothness as it is sensitive to the implicit ordering between states, may not yield the most appropriate discrete matrix for the original continuous relaxation, and does not guarantee constraint satisfaction. Despite notably reducing the distortion of the diffusion process, PRODIGY's fractional constraint enforcement does not address this drawback.
>
> In contrast, ConStruct operates directly on discrete state-spaces and avoids this limitation. ConStruct adjusts a graph from the previous reverse step by inserting the property preserving newly proposed edges, which are typically very few, promoting the process smoothness. Although edges are chosen in random order in the default implementation, Section H.3 empirically shows that this approach is not inferior to methods that order edge insertion based on predicted likelihoods by the diffusion model.
>
> **W4**: Theorem 1 provides useful guarantees about the proposed projector and helps understand the problem complexity. We agree that analyzing how often the projector retrieves the optimal graph is valuable; indeed, Theorem 2 shows that for the acyclicity constraint, the proposed projector *always* retrieves the optimal graph. Table 3 provides counter-examples demonstrating that this is not true for planarity, maximum degree, and lobster components. Since ConStruct generates new edges based on the diffusion model and not randomly, characterizing this subset to obtain more refined results (e.g., probabilistic bounds) for arbitrary properties is far from trivial. Finally, we believe that ConStruct is indeed well-suited for application as it shows significant performance improvements, high efficiency, and guaranteed property satisfaction.
>
> **W5**: We did not initially consider SPECTRE as closely related because it is GAN-based and it addresses unconstrained generation by producing eigenvectors and eigenvalues of the Laplacian and generating graphs conditioned on these. We did not find steps handling constraints, apart from the auxiliary eigenvector refinement matrices construction (must belong to the special orthogonal group). We would be happy to discuss SPECTRE’s relevance if the reviewer clarifies why it is particularly pertinent.
>
> **W6**: While ConStruct does not cover all PRODIGY constraints (e.g., molecular properties, see W7), it does handle a meaningful subset, including all of those tested on non-attributed datasets (edge count, triangle count, maximum degree). Crucially, ConStruct also handles combinatorial constraints (e.g., planarity, acyclicity), which PRODIGY cannot.
>
> **W7**: ConStruct does not handle node feature-dependent constraints found in molecular generation. However, molecular generation is dominated by autoregressive models due to their ability to perform validity checks at each step with minimal overhead [1]. The possibility of node ordering in molecules (via canonical smiles) explains this success. In contrast, ConStruct is designed for settings where node ordering is not suitable, such as digital pathology, focusing on purely structural constraints. While integrating node-dependent constraints into ConStruct is an exciting prospect, we believe that our focus on structural constraints does not diminish our contribution in a fundamentally distinct setting, as highlighted by Reviewer FDH7 in W1.
>
> **Limitations**:
> We appreciate the reviewer's suggestion and have elaborated on the specified points in the final manuscript (items 3 and 4 of global rebuttal). Overall, we believe we have addressed the limitations of our work adequately, a view also supported by Reviewer FDH7

---

> > ### Comment · Reviewer_ZwhZ · 2024-08-09
> >
> > Thanks for the rebuttal! However, I don't agree with the authors' views on certain topics and hope we can discuss and reach a conclusion.
> >
> > **Motivation:** The major problem I have with this is that the structural constraint of the underlying graphs must be known beforehand. As noted, this helps in the case of synthetic graphs which are specifically formed of such a constraint and maybe some generalizations can happen for specific use-cases as identified by authors for pathology graphs and planarity. However, the major benefit of graph generation is that it can approximate a distribution just from data. What this paper proposes is to include an additional constraint that identifies the underlying distribution. This is good if the proposed method is motivated as a way to simulate synthetic NP-hard structural distributions, which takes time to generate otherwise. However, in the absence of such a clear motivation and an expectation to generalize to real-world graphs, it is not clear how the users would be able to identify structural constraints of arbitrary graph distributions before training.
> >
> > **Comparison with SPECTRE:** SPECTRE trains generative model by explicitly conditioning the eigenspectrum of the training data. This explicit conditioning on the underlying distribution is similar to the idea of explicitly constraining on the structural constraint of planarity and acyclicity that the current paper looks at. For example, if we have graphs with 2 connected components, it can be inferred from the number of non-zero eigenvalues in the training graphs. However, this discussion is absent in the current work but is extremely important.
> >
> > **Comparison with PRODIGY:** This paper also requires further discussion that the authors have not acknowledged. The difference is not in continuous vs discrete or combinatorial vs not. PRODIGY constraints are also inherently combinatorial as opposed to what the authors are claiming. They have particularly considered P-space matroid constraints while ConStruct focuses on NP-hard constraints. However, more importantly, the difference is that ConStruct is a discrete diffusion model trained to generate graphs given a structural constraint on the distribution while PRODIGY aims to do plug-and-play controllable graph generation to satisfy arbitrary constraints. The authors have been inconsistent in claiming (with the guidance comment) that they may be doing the same problem as PRODIGY, then such an experimental validation would be essential to show.
> >
> > **Contribution of the method:** The theorems that assist the proposed method are not quite generalizable. Theorem 1 barely shows that a satisfiable constrained graph can be sampled, which is also possible through a simple random sampling. Theorem 2, on the other hand, is specific to a single constraint. I agree that any such theoretical proof will be extremely hard to prove for a general graph constraint. However, in that case, the authors should not overclaim their contributions in terms of constraint satisfaction with discrete graphs. The experiments are strong enough to validate the method's usefulness.
> >
> > **Attributed settings:** I am not diminishing the contributions but I would like to keep my opinion that this is a major weakness in using the method in real-world settings.

---

> > > ### Author Response · Authors · 2024-08-12
> > >
> > > We thank the reviewer for the detailed reply to our rebuttal. We address the raised concerns below:
> > >
> > > **Motivation**: We agree with the reviewer that the flexibility provided by the unsconstrained graph generation of mimicking any data distribution is extremely convenient.
> > > However, in many real-world scenarios, this flexibility alone is not sufficient to yield satisfactory performance without incorporating additional priors into the generative model.
> > > This approach is particularly relevant in data scarce settings — such as due to high costs, ethical/privacy concerns, or a lack of high-quality annotated data — or where instances that do not adhere to specified constraints are either infeasible or lack physical meaning.
> > > Constrained generation is a valuable method for incorporating such priors by hard-constraining the hypothesis space to valid instances (in our cases, graphs), thereby reducing the search space and potentially enhancing the efficacy of the learning process.
> > > We note, nevertheless, that just enforcing such constraint does not identify the underlying distribution; instead, there is still the need to learn the underlying distribution within the constrained domain.
> > >
> > > As outlined in our rebuttal (W1), constrained generation indeed requires an explicit and unambiguous definition of the constraints. We understand the reviewer’s concerns about identifying structural constraints *a priori*, but we do not view this as a limitation. Instead, constrained generation is a practical choice for settings where practitioners have established explicit constraints through expertise or problem exposure and wish to leverage them for a more effective learning process. Typical sources of such constraints include the physics of the problem, application domain constraints, or detailed knowledge about the data acquisition process. We highlight several real-world examples (some mentioned in the paper) where such prior knowledge is available and can be effectively applied using ConStruct, showcasing its versatility.
> > >
> > > - *Planarity*: design of road networks [1], chip design [2], biochemistry [3] or digital pathology [4].
> > >
> > > - *Acyclicity*: evolutionary biology [5] or epidemiology [6]. Additionally, if we consider the extension of discrete diffusion for directed graphs, e.g. [7], for which ConStruct is still applicable, there are several domains where the generation of directed acyclic graphs is critical: neural architecture search or bayesian network structure learning [8], causal discovery [9], etc.
> > >
> > > - *Maximum Degree*: design of contact networks [10, 11].
> > >
> > > **Comparison to SPECTRE**:
> > > Even though we still see some relevant differences between ConStruct and SPECTRE - such as, 1) ConStruct enforces constraints by default, while SPECTRE requires learning such dependencies even with explicit structural information; or 2) SPECTRE uses spectral properties to inform the model about global graph structure, whereas ConStruct can still be used to constrain generation for local level properties (e.g., maximum degree) -, we agree with the reviewer that the latter is the first method, to the best of our knowledge, that recognizes the importance of explicitly incorporating structural information (other than locality biases from common GNNs) as powerful priors for the expressiveness of one-shot graph generative models. Therefore, we will add this discussion into our final version of the manuscript. We thank the reviewer for the constructive feedback.

---

> > > > ### Author Response · Authors · 2024-08-12
> > > >
> > > > **Comparison with PRODIGY**: We agree with the reviewer that the two methods can be used to address different sets of constraints and, more importantly, that the main difference between PRODIGY and ConStruct consists of the different tasks addressed by both, as accurately described by the reviewer. The fact that both tasks are different lead to fundamental differences in the two approaches. For example, PRODIGY does not have to necessarily guarantee constraint satisfaction, whilst ConStruct does.
> > > >
> > > > The comment regarding guidance points towards our section G.2. There, we use the projector to control the generation of an unconstrained model (i.e., trained with a noise model that does not preserve the structural constraint across the forward process). This approach is similar to PRODIGY's procedure but uses our projector instead. Even though this approach accomplishes satisfactory performance, it is significantly outperformed by a ConStruct model (where the noise model preserves the constraint) in terms of validity: 81.3\% vs 99.8\%. As discussed in that section, this experiment is meaningful to reinforce the foundation upon which ConStruct is designed: matching the forward and reverse domains allows for in distribution prediction of the denoising neural network, improving constrained generation performance. We apologize if this reference was misleading.
> > > >
> > > > To address the reviewer's concerns, we propose to extend our paragraph dedicated to PRODIGY of our related work section (ll. 80-87, page 3) to make a more explicit distinction between both methods on the different nature of constraints addressed and, specially, on the different tasks tackled by each of them.
> > > >
> > > > **Contribution of the method**: We thank the reviewer for acknowledging the suitability of our method for application. Our intention with the provided theoretical analysis is to provide insights to the reader that underscore the complexity of the problem and substantiate the design of our projector. For instance, Theorem 1 implies that a deterministic projector based on edge sampling likelihoods from the diffusion model is not guaranteed to have the optimal graph in its output set, as elaborated in Appendix H.3. Also, Theorems 1 and 2 and the counter-examples in Figure 3 show that the problem that the projector is addressing is not so trivial such that it can always output the optimal graph in the GED sense for all the edge-deletion invariant properties. Nevertheless, our projector is *guaranteed* to produce graphs that meet the specified structural constraints. Considering this, we remain open to discussing and refining any specific claims in our manuscript that might be seen as overstated by the reviewer.
> > > >
> > > > **Attributed settings**: We appreciate the reviewer’s opinion but do not fully understand why the absence of node feature-dependent constraints is seen as a *major* weakness. As argued in W7 of rebuttal, ConStruct targets applications where node ordering is not feasible. In contrast, node-edge joint constraints are primarily useful for constrained molecular generation, a domain where node ordering is possible and thus perfectly suited for autoregressive models. Due to their ability to perform very efficient validity checks at each step, these models already successfully solve such task. While integrating node-edge constraints is an interesting potential extension to ConStruct, we believe that our focus on structural constraints adresses our intended applications and does not constitute a fundamental weakness.
> > > >
> > > > ____
> > > >
> > > > We believe that the constructive feedback provided by the reviewer has guided us toward addressing their concerns more effectively. In light of these improvements, we hope the reviewer might consider revising their score.

---

> > > > > ### Author Response · Authors · 2024-08-12
> > > > >
> > > > > [1] - Xie et al. "Topological evolution of surface transportation networks." Computers, Environment and Urban Systems, 2009.
> > > > >
> > > > > [2] - Bhatt et al. "A framework for solving VLSI graph layout problems." Journal of Computer and System Sciences, 1984.
> > > > >
> > > > > [3] - Simmons III et al. "Synthesis of the first topologically non-planar molecule." Tetrahedron Letters, 1981.
> > > > >
> > > > > [4] - Jaume et al. "Histocartography: A toolkit for graph analytics in digital pathology." MICCAI Workshop on Computational Pathology. PMLR, 2021.
> > > > >
> > > > > [5] - Gregory. "Understanding evolutionary trees." Evolution: Education and Outreach, 2008.
> > > > >
> > > > > [6] - Seibold et al. "Modeling epidemics on a regular tree graph." Letters in Biomathematics, 2016.
> > > > >
> > > > > [7] - Asthana, Rohan, et al. "Multi-conditioned Graph Diffusion for Neural Architecture Search." arXiv, 2024.
> > > > >
> > > > > [8] - Zhang, Muhan, et al. "D-vae: A variational autoencoder for directed acyclic graphs." NeurIPS, 2019.
> > > > >
> > > > > [9] - Sanchez et al. "Diffusion models for causal discovery via topological ordering." arXiv, 2022.
> > > > >
> > > > > [10] - Jang et al. "Evaluating architectural changes to alter pathogen dynamics in a dialysis unit: for the CDC MInD-healthcare group." International Conference on Advances in Social Networks Analysis and Mining, 2019.
> > > > >
> > > > > [11] - Kong et al. "Autoregressive diffusion model for graph generation." ICML, 2023.

---

> > > > > > ### Comment · Reviewer_ZwhZ · 2024-08-13
> > > > > >
> > > > > > I thank the authors for providing additional discussion and clarifications regarding their work that positions their paper better in my opinion. I believe they should thus include all these points in their revised version. I don't think non-attributed settings is a major weakness but worth mentioning as a limitation and potential for future research. I will raise my scores now that I think we are on the same page.

---

> > > > > > > ### Author Response · Authors · 2024-08-14
> > > > > > >
> > > > > > > We thank the reviewer for their time, constructive feedback, and the updated score. We will incorporate the additional discussions and clarifications into the revised version of the paper.

---

> ### Author Response · Authors · 2024-08-06
>
> [1] - Mercado, Rocío, et al. "Graph networks for molecular design." Machine Learning: Science and Technology 2021

---

### Official Review · Reviewer_wnSX · 2024-07-14

**Soundness:** 3
**Presentation:** 3
**Contribution:** 3
**Rating:** 7
**Confidence:** 4

**Summary:**

This paper presents ConStruct, a framework incorporating structural constraints into graph generation models using a discrete graph diffusion process. By introducing an edge-absorbing noise model and a projector operator, ConStruct ensures generated graphs meet specific properties like planarity or acyclicity, crucial for real-world applications. The framework significantly improves the validity of generated graphs, demonstrated through experiments on synthetic benchmarks and real-world datasets such as digital pathology, achieving up to a 71.1 percentage point increase in graph validity. It mainly focuses on addressing the challenge of integrating domain knowledge into graph generative models, enhancing their practical deployment.

**Strengths:**

1. Maintaining the structural constraint of the generated graph is an important but challenging problem for diffusion-based graph generative models.

2. The design of the edge-absorbing noise model and projector operator is reasonable and technically sound. Especially the efficiency consideration of incremental validity checks.

3. The experimental results are good compared with recent SOTA baselines, including diffusion-based methods. The extensive results provided in appendix also well support the advantage of proposed method.

4. The presented evaluation on Digital Pathology Graph dataset is interesting, and the released dataset seems useful for future research.

**Weaknesses:**

1. As explained by the authors, the proposed method can only deal with a specific type of structural constraint, i.e., edge-deletion invariance, but the discussion on the possible impact of this limitation is not well clarified. For example, what other normally-seen constraint

2. Lack of complexity analysis. Instead, only the runtime measurement is provided.

3. This method might not be able to deal with large graphs.

**Questions:**

Please answer my listed weakness above.

**Limitations:**

None.

---

> ### Author Rebuttal · Authors · 2024-08-06
>
> We sincerely appreciate the reviewer’s constructive feedback and requests for clarification. We have addressed the raised questions below:
>
> **W1**: An example of other purely structural constraints that ConStruct does not cover by default are edge-insertion invariant properties, *i.e.*, properties that hold upon edge insertion. This type of constraint can be useful, for instance, in the molecular generation domain, where we could impose constraints like "the generated graph should at least have a cycle" (we explore the inverse setting — acyclic molecular generation — in Appendix G.2). We note, nevertheless, that edge-insertion invariant properties could be easily captured by our framework through two simple modifications: design the transition matrices with the absorbing state in the existing edge state (instead of in the no-edge state) and a projector that progressively removes edges (instead of inserting them) while preserving the desired property. This simple inversion of ConStruct would allow it to cover edge-insertion invariant properties. In item 3 of the global rebuttal, we discuss other meaningful constraints that ConStruct does not cover but that could be extended to as future work, even if not purely structural. We propose to explicitly add this discussion to the paper.
>
> Additionally, an example of a structural constraint that is completely outside the scope of ConStruct can be seen in graph stochastic models. Since the validity of these types of graphs is determined via statistical tests (thus, not deterministically), it is difficult to constrain the generation in that setting. An example of this occurs in the graph generation benchmark SBM dataset (from Stochastic Block Model), where validity is checked using a Wald test [1]. To the best of our knowledge, constrained diffusion towards such graph structural properties remains an open problem.
>
> **W2**: Since the projector is the only component incurring overhead, ConStruct only interferes with the complexity of the sampling algorithm, leaving the efficiency of the training algorithm intact. As described in Section 3.5, through the use of a blocking edge hash table and incremental property satisfaction algorithms, we are able to minimize the increase in complexity imposed by the projector. The complexity analysis for the blocking edge hash table is provided directly in Section 3.5. The complexity analysis for all the different edge-deletion invariant properties explored in the paper (planarity, acyclicity, lobster components, and maximum degree) can be found in Section 3.5 (planarity) and in Appendix C (all of them). We also provide a more explicit comparison between the underlying discrete diffusion algorithm and the incurred overhead by the projector in item 1 of the global rebuttal. We remark that due to the combinatorial nature of edge-deletion invariant properties, each property satisfaction algorithm is specific to the property at hand. Therefore, there is no general efficient property satisfaction algorithm for all edge-deletion invariant properties, and we have to address each property on a case-by-case basis.
>
> **W3**: Item 1 of the global rebuttal details the minor overhead imposed by the projector within the diffusion framework. It also addresses the method’s (high) scalability relative to the underlying discrete diffusion framework. In particular, we demonstrate that as the generated graph size increases, the overhead of ConStruct becomes increasingly negligible compared to the underlying discrete diffusion framework. Therefore, as far as that discrete diffusion framework is able to scale, ConStruct remains a viable option.
>
> [1] - Martinkus, Karolis, et al. "Spectre: Spectral conditioning helps to overcome the expressivity limits of one-shot graph generators." ICML, 2022.

---

### Author Rebuttal · Authors · 2024-08-06

# Global Rebuttal

We kindly thank all the reviewers for their time and valuable feedback on our work.

As a brief overview, our paper presents ConStruct, the first graph constrained diffusion framework to fully run in the discrete setting. ConStruct guarantees the satisfaction of edge-deletion invariant constraints through the application of a specific edge-absorbing noise model and a new projector operator. These components ensure matched training and sampling distributions, thereby improving performance over unconstrained models in synthetic datasets and real-world applications.

___

In response to the reviewers' comments, we have updated our paper to include a series of clarifications and minor corrections to enhance the understanding of our work. We enumerate these updates below:

1) **Complexity Comparison**: in response to reviewers wnSX and FDH7, we provide a more explicit comparison between the underlying diffusion model and the overhead incurred by the projector in terms of complexity:

    "At each reverse step (out of a total of $T \approx 1000$ steps), the denoising network makes predictions for all nodes and pairs of nodes. This results in $O(n^2)$ predictions per step. Thus, the complexity of the sampling algorithm of the underlying discrete diffusion model is $O(n^2 T)$. In addition, the complexity overhead imposed by the projector is $O(N V)$. Here, $V$ represents the complexity of the property satisfaction algorithm, and $N$ is the number of times this algorithm is applied. So, in total, we have $O(n^2T + NV)$.

    Our analysis in Appendix C shows that incremental property satisfaction algorithms have notably low complexity. For instance, in cases like acyclicity, lobster components, and maximum degree, we have $V=O(|E_\text{added}|)$. Since the projector adds one edge at a time, we have $V=O(1)$. Additionally, since the blocking edge hash table limits us to perform at most one property satisfaction check per newly proposed edge (either we have never tested it or it is already blocked), $N$ corresponds to the total number of different edges proposed by the diffusion model across the whole reverse process. Thus, we necessarily have $N \leq n^2$. For these reasons, we directly find $O(N V) \ll O(n^2 T)$, highlighting the minimal overhead imposed by the projector compared to the discrete diffusion model. This explains the low runtime overhead observed for ConStruct, as detailed in Section D.3 (9\% for graphs of the tested size).

    A reasonable assumption is that the model inserts $N = O(|E|)$ edges throughout the reverse process. This is for example true if the model is well trained and predicts the correct graph. Besides, most families of graphs are sparse, meaning that $\frac{|E|}{n^2} \to 0$ as $n\to \infty$. For example, planar and tree graphs have to satisfy $|E| / n^2 = O(1/n)$. Therefore, we can conclude that asymptotically $O(n^2T + NV) = O(n^2T)$, *i.e.*, the projector overhead becomes increasingly negligible relative to the diffusion algorithm itself as the graph size increases. This further highlights the scalability of our proposed method."

2) **Notation and Citations**: in response to reviewer FDH7, we add the definition of $\Delta^k$ as the $k$-simplex to the notation paragraph of the paper and we updated our citations of GraphARM and EDGE to their published version;

3) **Limitations on Constraints**: in response to all reviewers, we provide more information regarding the limitation of ConStruct to edge-deletion invariant constraints:

    "In our work, we cover edge-deletion invariant properties. However, ConStruct can be easily extended to also handle edge-insertion invariant properties (*i.e.*, properties that hold upon edge insertion). This extension is particularly useful in domains where constraints such as having at least $n$ cycles in a graph can be important. To achieve this, we can simply "invert" the proposed framework: design the transition matrices with the absorbing state in an existing edge state (instead of the no-edge state) and a projector that removes edges progressively (instead of adding them) while conserving the desired property.

    In the particular context of molecular generation, Appendix G illustrates that, while purely structural constraints can guide the generation of molecules with specific structural properties (e.g., acyclicity), for general properties shared by all molecules (e.g., planarity) they are too loose. In constrast, autoregressive models thrive in such setting due to the possibility of molecular node ordering (e.g., via canonical smiles) and the efficient incorporation of *joint node-edge* constraints. Therefore, even though it consists of a fundamentally different setting than the one considered in this paper, incorporating joint node-edge constraints into ConStruct represents an exciting future direction for our work."

4) **Impact on digital Pathology**: in response to reviewer ZwhZ, we extend the already existing impact statement (appendix J) with a more detailed stance in the potential impact of our work for the particular setting of digital pathology:

    "For the particular case of the digital pathology setting, while the generated graphs are able to mimic clinically relevant structures, they remain too small to have any direct clinical impact. Pathologists use whole-slide images for informed decisions, whose corresponding cell graphs typically comprise a total number of nodes 3 to 4 orders of magnitude above the graphs generated at this stage."

    If deemed appropriate by the reviewers, we are also considering moving Appendix J to the main body of the paper.
___
We reiterate our appreciation for the valuable feedback provided by the reviewers and hope that the updated version of the paper, along with the individual replies to each reviewer, have addressed the main concerns raised. We remain open to any further discussion.

---

### Author Response · Authors · 2024-08-14
**Appreciation for Your Efforts and Feedback**

Dear Reviewers and Area Chairs,

Thank you for managing our submission. We are pleased to see that our rebuttal successfully addressed the reviewers' concerns. We greatly value the feedback provided, which has clearly enhanced the quality of our paper.

If there are any remaining issues or additional points that need further attention, please let us know. We really appreciate your time and consideration.

Best regards,

The Authors

---

### Decision · Program_Chairs · 2024-09-25

**Decision:**

Accept (poster)

**Comment:**

This paper introduces ConStruct, a novel framework that incorporates structural constraints into graph generation models using a discrete graph diffusion process. By employing an edge-absorbing noise model and a projector operator, ConStruct ensures that generated graphs adhere to specific properties, such as planarity or acyclicity, which are crucial for practical applications. The framework demonstrates a significant improvement in the validity of generated graphs, as evidenced by experiments on both synthetic benchmarks and real-world datasets, such as digital pathology, showing a substantial increase in graph validity.

The reviewers unanimously recommend that this manuscript be accepted. However, they also suggest including some relevant baselines and conducting a theoretical analysis of the sample complexity. The authors are advised to incorporate the reviewers’ feedback when preparing the final version of the paper.